
# WETMETH 1.0: A new wetland methane model for implementation in Earth system models

Claude-Michel Nzotungicimpaye[1], Andrew H. MacDougall[2], Joe R. Melton[3], Claire C. Treat[4], Michael Eby[5], Lance F.W. Lesack[1, 6], Kirsten Zickfeld[1]

[1]Department of Geography, Simon Fraser University, Burnaby, BC, Canada
[2]Climate and Environment, St. Francis Xavier University, Antigonish, NS, Canada
[3]Climate Research Division, Environment and Climate Change Canada, Victoria, BC, Canada
[4]Alfred Wegener Institute Helmholtz Centre for Polar and Marine Research, Potsdam, Germany
[5]School of Earth and Ocean Sciences, University of Victoria, Victoria, BC, Canada
[6]Department of Biological Sciences, Simon Fraser University, Burnaby, BC, Canada

*Correspondence to*: Claude-Michel Nzotungicimpaye (cnzotung@sfu.ca)

**Abstract.** Wetlands are the single largest natural source of methane ($CH_4$), a powerful greenhouse gas affecting the global climate. In turn, wetland $CH_4$ emissions are sensitive to changes in climate conditions such as temperature and precipitation shifts. However, biogeochemical processes regulating wetland $CH_4$ emissions (namely microbial production and oxidation of $CH_4$) are not routinely included in fully coupled Earth system models that simulate feedbacks between the physical climate, the carbon cycle, and other biogeochemical cycles. This paper introduces a process-based wetland $CH_4$ model (WETMETH) developed for implementation in Earth system models and currently embedded in an Earth system model of intermediate complexity. Here we: (i) describe the wetland $CH_4$ model; (ii) evaluate the model performance against available datasets and estimates from the literature; (iii) analyze the model sensitivity to perturbations of poorly constrained parameters. Historical simulations show that WETMETH is capable of reproducing mean annual emissions consistent with present-day estimates across spatial scales. For the 2008-2017 decade the model simulates global mean wetland emissions of 158.6 Tg $CH_4$ yr$^{-1}$, of which 33.1 Tg $CH_4$ yr$^{-1}$ are from wetlands north of 45°N. WETMETH is highly sensitive to parameters for the microbial oxidation of $CH_4$, which is the least constrained process in the literature.





# 1 Introduction

Wetlands are vegetated locations that are inundated with water on a permanent, seasonal or recurrent basis (Wheeler, 1999). In the context of this study, wetlands are defined following the latest global $CH_4$ budget report (Saunois et al., 2020): natural ecosystems with inundated or water-saturated soils where anoxic conditions lead to the production of $CH_4$. Wetlands across the globe are the single largest natural source of atmospheric $CH_4$, accounting for approximately a third of total global emissions (Bridgham et al., 2013; Saunois et al., 2016). Estimates of global wetland $CH_4$ emissions over the past few decades vary between 140 and 210 Tg $CH_4$ yr$^{-1}$ (Kirschke et al., 2013). Although there exist different types of wetlands such as bogs, fens, swamps, marshes and floodplains (Aselmann and Crutzen, 1989; Saunois et al., 2016), the release of $CH_4$ from any wetland results from the balance between two biogeochemical processes (Segers, 1998): the production of $CH_4$ by anaerobic microbes (namely methanogens) and the oxidation of $CH_4$ primarily by aerobic microbes (namely methanotrophs).

Both $CH_4$ production and oxidation in wetlands are sensitive to changes in climate conditions. For instance, soil warming accelerates the microbial activity with a higher response for methanogenic than methanotrophic activity (Bridgham et al., 2013; Dunfield et al., 1993; Segers, 1998). At the landscape or larger scale, increased wet conditions tend to enhance methanogenic activity to the detriment of methanotrophic activity (Duval and Radu, 2018; Helbig et al., 2017; Kim, 2015). In turn, wetland $CH_4$ emissions can affect the global climate through changes in atmospheric $CH_4$ levels and associated radiative forcing (Dean et al., 2018; O'Connor et al., 2010). Analyses of ice cores suggest that $CH_4$ emissions from tropical and northern wetlands contributed significantly to climate changes during past glacial-interglacial transitions (Loulergue et al., 2008; Rhodes et al., 2017).

The interactions between climate conditions and wetland $CH_4$ emissions translate into a positive feedback loop that has the potential to amplify changes in global mean surface air temperature, which is a major concern for future climates (Dean et al., 2018; O'Connor et al., 2010). Research on feedbacks between the physical climate and biogeochemical cycles is generally conducted with 3-dimensional (3-D) fully coupled Earth system models (ESMs) (Arora et al., 2013). Over the past decade, these ESMs have proven very useful to investigate and inform international climate policies such as the accounting of carbon emissions required to avoid the risk of dangerous climate change (Zickfeld et al., 2009) and achieve the goals of the Paris Agreement (Tokarska and Gillett, 2018). Yet, biogeochemical processes regulating $CH_4$ emissions in wetlands are not commonly included in fully coupled ESM simulations.

In the past, several process-based models have been developed for investigating the response of wetland $CH_4$ emissions to climate variability and climate change (Hodson et al., 2011; Hopcroft et al., 2011; Pandey et al., 2017; Paudel et al., 2016; Shindell et al., 2004; Zhang et al., 2018; Zhu et al., 2015). These wetland $CH_4$ models are generally embedded in terrestrial or land surface models and forced with observational datasets or reanalysis products (Melton et al., 2013; Wania et al., 2013; Xu et al., 2016). A second application for wetland $CH_4$ models has been to quantify the climate response to wetland $CH_4$ emissions (Gedney et al., 2004, 2019; Zhang et al., 2017b). In this case, results from wetland $CH_4$ models are used in climate-carbon cycle model emulators to assess their impact on radiative forcing (Gedney et al., 2019; Zhang et al.,



2017b). These modelling studies have contributed to advance research on the possible evolution of wetland $CH_4$ emissions in the 21$^{st}$ century (Koven et al., 2011; Shindell et al., 2004), the magnitude of their impact on the global climate (Gedney et al., 2019; Zhang et al., 2017b), and their implications for international climate policy (Comyn-Platt et al., 2018). However, their quasi-coupling methods do not reflect the complete feedback loop between climate conditions and wetland $CH_4$ emissions as expected in the natural world. So far, only 1-D and 2-D models of the northern high-latitude regions have been applied for simulating the feedback between climate conditions (temperature changes) and wetland $CH_4$ emissions in a fully coupled mode (Schneider von Deimling et al., 2012, 2015).

The implementation of process-based wetland $CH_4$ models in fully coupled ESMs is needed in order to advance research on wetland $CH_4$-climate feedbacks in the context of global climate projections (Dean et al., 2018). In particular, this addition to Earth system modelling should be beneficial to ongoing research on the permafrost carbon feedback (Nzotungicimpaye and Zickfeld, 2017; Schuur et al., 2015) and the remaining carbon budget for achieving the goals of the Paris Agreement (Rogelj et al., 2019).

This paper introduces a wetland $CH_4$ model developed for implementation in ESMs and currently embedded in an Earth system model of intermediate complexity (EMIC). Our study aims at developing a computationally efficient process-based model for simulating large-scale wetland $CH_4$ emissions constrained with sparse observations. Section 2 gives an overview of processes regulating $CH_4$ emissions in wetlands. Section 3 provides the model description and an outline of performed model simulations. Section 4 describes the model calibration and choice of parameter values. Section 5 presents the model performance evaluation. Section 6 describes the model sensitivity to poorly constrained parameters. Sections 7 and 8 are for discussions and conclusions, respectively.

## 2 Overview of processes regulating methane emissions in wetlands

### 2.1 Microbial production of methane

Wetlands host several communities of microbes adapted to the predominant anoxic conditions of these environments (Bridgham et al., 2013). Some of these microbes are methanogens, which decompose organic matter for their metabolism and produce $CH_4$ as a by-product of their respiration (McCalley et al., 2014; Segers, 1998). The organic matter decomposed by methanogens in wetlands originates from litter-fall, root exudates, dead plants and dissolved organic carbon (Bridgham et al., 2013; Conrad, 2009; Girkin et al., 2018; Mitsch and Mander, 2018). In the northern permafrost region, carbon from thawed soils constitutes an additional source of organic matter to methanogens (Kwon et al., 2019; Olefeldt et al., 2013).

There are three pathways through which methanogens produce $CH_4$ from soil organic matter (Le Mer and Roger, 2001; Segers, 1998; Whalen, 2005). The first pathway is operated by methanogens that rely on acetate for their metabolism, resulting in the production of both $CH_4$ and carbon dioxide ($CO_2$) (Bridgham et al., 2013; Whalen, 2005). The second pathway is operated by methanogens that produce $CH_4$ through $CO_2$ reduction in the presence of hydrogen (Bridgham et al.,





2013). The third pathway is operated by methanogens that use methylated substrates (e.g. methanol, methylamines, and
dimethysulfide) for their metabolism (Zalman et al., 2018).

Rates of $CH_4$ production in wetlands are generally highest in upper anoxic layers due to several factors such as the quality of organic matter and the spread of active microbial populations. For instance, in comparison to soil layers at depth where organic matter can be recalcitrant to microbial decomposition, the organic matter in near-surface soil layers is more labile due to fresh inputs from litter-fall and vegetation mortality (Treat et al., 2015; Walz et al., 2017; Wild et al., 2016).
Furthermore, observations at various sites show that methanogenic activity decreases as depth increases (Bridgham et al., 2013; Cadillo-Quiroz et al., 2006).

Increasing soil temperatures stimulate the dynamics and growth of methanogenic communities in wetlands, resulting in an increase of $CH_4$ production rates (Bridgham et al., 2013; Segers, 1998). However, several studies indicate that there is an optimal temperature for methanogenic activity between 25°C and 30°C (Dean et al., 2018; Dunfield et al., 1993).
Other factors promoting the occurrence of $CH_4$ production in wetlands include the persistence of anoxic conditions as well as soil pH varying between acidic and neutral (Dunfield et al., 1993; Segers, 1998).

## 2.2 Microbial oxidation of methane

In wetlands, methanotrophs ($CH_4$-oxidizing microbes) populate oxic portions of the soil column (Bridgham et al., 2013; Conrad, 2009; Whalen, 2005). Such oxic portions are primarily soil layers close to the surface which are in contact with the
atmosphere, commonly near and above the water table (Bridgham et al., 2013; Le Mer and Roger, 2001; Segers, 1998). In the presence of vascular plants, other oxic portions of the soil column can be found near the roots due to the downward transport of oxygen ($O_2$) through plant aerenchyma (Kwon et al., 2019; Whalen, 2005). All these oxic portions of the soil column constitute the so-called oxic zone, which is predominantly made of soil layers near and above the water table (Bridgham et al., 2013; Conrad, 2009; Segers, 1998). Methanotrophs consume $CH_4$ that ascends from the zones of
production at depth to the overlying oxic zone for their metabolism, and primarily produce $CO_2$ as part of their respiration (Bridgham et al., 2013; Segers, 1998).

While $O_2$ has been considered for years to be the only electron acceptor involved in the microbial oxidation of $CH_4$, there is a growing evidence of the occurrence of $CH_4$ oxidation under anoxic conditions operated by anaerobic microbes that rely on alternate electron acceptors such as nitrate and sulfate (Dean et al., 2018). However, although anaerobic $CH_4$
oxidation in marine environments has been well established for decades (Hoehler et al., 1994; Reeburgh, 1976), this process remains poorly investigated in wetlands despite its potential importance for the $CH_4$ cycle (Gauthier et al., 2015; Smemo and Yavitt, 2011).

In analogy to $CH_4$ production, $CH_4$ oxidation is influenced by changes in soil temperatures (Bridgham et al., 2013; Segers, 1998). For instance, $CH_4$ oxidation rates increase during the summer because of intensified microbial activity but
also the availability of substantial $CH_4$ in response to increased soil temperatures (Segers, 1998). However, the temperature



response for $CH_4$ oxidation is generally lower than that for $CH_4$ production (Bridgham et al., 2013; Dean et al., 2018; Dunfield et al., 1993; Segers, 1998).

## 2.3 Mechanisms transporting methane to the atmosphere

There exist various mechanisms transporting $CH_4$ produced in wetlands to the atmosphere. Three transport mechanisms are
well documented in the literature and generally monitored in situ (Bridgham et al., 2013; Whalen, 2005): the diffusion of $CH_4$ whereby molecules of $CH_4$ slowly ascend the overlying water column, the ebullition of $CH_4$ whereby bubbles of $CH_4$ rapidly ascend towards the soil surface, as well as the transport of $CH_4$ through the aerenchyma of vascular plants. However, other transport mechanisms for $CH_4$ in wetlands have been revealed: the hydrodynamic transport of $CH_4$ in the form of upwelling caused by temperature gradients primarily at nighttime (Poindexter et al., 2016), and the transport of $CH_4$ through
tree stems (Bridgham et al., 2013; Conrad, 2009; Pangala et al., 2017) whose driving processes are still not well understood (Barba et al., 2019).

Methane oxidation is highly dependent on the predominant transport mechanism for $CH_4$. The water table position plays a crucial role in affecting what fraction of the produced $CH_4$ reaches the atmosphere (Blodau, 2002; Moore and Roulet, 1993; Segers, 1998). When the water table is well below the surface, methanotrophs may oxidize all of the diffusing $CH_4$
before the gas reaches the atmosphere (Segers, 1998). In the presence of vascular plants, a lower fraction of the produced $CH_4$ is oxidized because these plants allow the gas to bypass the oxic zone where methanotrophs are hosted (Blodau, 2002; Bridgham et al., 2013; Segers, 1998). In the case of ebullition, which often occurs episodically, $CH_4$ may escape to the atmosphere with reduced opportunities for oxidation (Bridgham et al., 2013; Whalen, 2005). How $CH_4$ oxidation relates to the transport of $CH_4$ through tree stems (Barba et al., 2019) or by hydrodynamic processes (Poindexter et al., 2016) is not
well established.

## 2.4 A synopsis of wetland methane dynamics

Fig. 1 illustrates vertical profiles of soil organic content, $CH_4$ concentration, and $CH_4$ oxidation rates in a soil column with and without inundation at the surface based on principles outlined in the literature (Blodau et al., 2004; Whiticar and Faber, 1985). In general, the water table position determines the maximum depth at which $O_2$ is available in the soil column (i.e. the
oxic-anoxic interface). When the surface is flooded and the water is stagnant (Fig. 1a), $O_2$ diffuses slowly into the soil column and may only be present in a portion of the upper soil layer which is in contact with the atmosphere. Under such predominantly anoxic conditions, $CH_4$ production occurs throughout the soil column and the concentration of $CH_4$ mirrors soil organic content – eventually with a small reduction near the surface due to $CH_4$ oxidation. A modest amount of ascending $CH_4$ may be oxidized throughout the soil column, but with highest oxidation rates near the surface where some $O_2$
may be available as an electron acceptor. The combination of high $CH_4$ production and only modest $CH_4$ oxidation in the soil column results in large $CH_4$ emissions into the atmosphere.



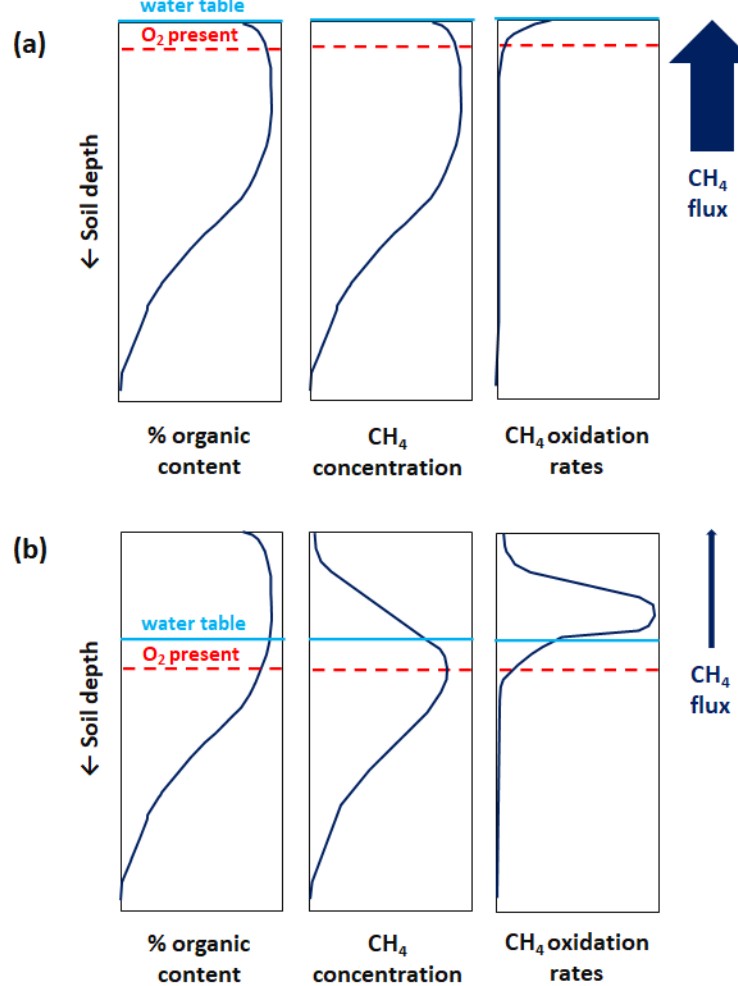

**Figure 1: Illustrated vertical profiles of soil organic content, CH₄ concentration and oxidation rates in a soil column with inundation at the surface (a) and without inundation at the surface (b). The vertical profiles are based on principles outlined in the literature (Blodau et al., 2004; Whiticar and Faber, 1985). For simplicity, the soil organic content is assumed to be identical in (a) and (b). In each case, the blue horizontal line illustrates the water table position and the dashed red horizontal line illustrates the oxic-anoxic interface or maximum depth at which O₂ is available in the soil column. The relative magnitude of CH₄ flux in the soil column is shown by the upward arrow to the right, also characterizing the relative magnitude of CH₄ emissions into the atmosphere.**

When the flooding recedes, $O_2$ becomes more prevalent in the upper soil column where $CH_4$ concentration decreases following a slow down or shut down of $CH_4$ production as aerobic microbes dominate the competition for organic matter (Fig. 1b). $CH_4$ production persists below the oxic-anoxic interface where the concentration of $CH_4$ mirrors soil organic content owing to the predominant anoxic conditions. Ascending $CH_4$ becomes subject to substantial oxidation in the soil column with the highest oxidation rates above the oxic-anoxic interface where $O_2$ is abundant. The combination of decreased $CH_4$ production and substantial $CH_4$ oxidation in the soil column results in small or no $CH_4$ emissions into the atmosphere.



## 3 Model description and simulations

### 3.1 The wetland methane model: WETMETH

Microbial production and oxidation of $CH_4$ are parameterized in WETMETH using a multi-layer ground structure with
information on the moisture distribution, the amount of organic matter (carbon content), and the average temperature in each soil layer. These soil variables are commonly simulated by ESMs. Fig. 2 provides a schematic representation of WETMETH for a soil column with and without inundation at the surface. By configuration, it is considered that $CH_4$ emissions in WETMETH may occur not only from inundated locations, but also from non-inundated ecosystems with a relatively high level of soil moisture content (Saunois et al., 2016, 2020).

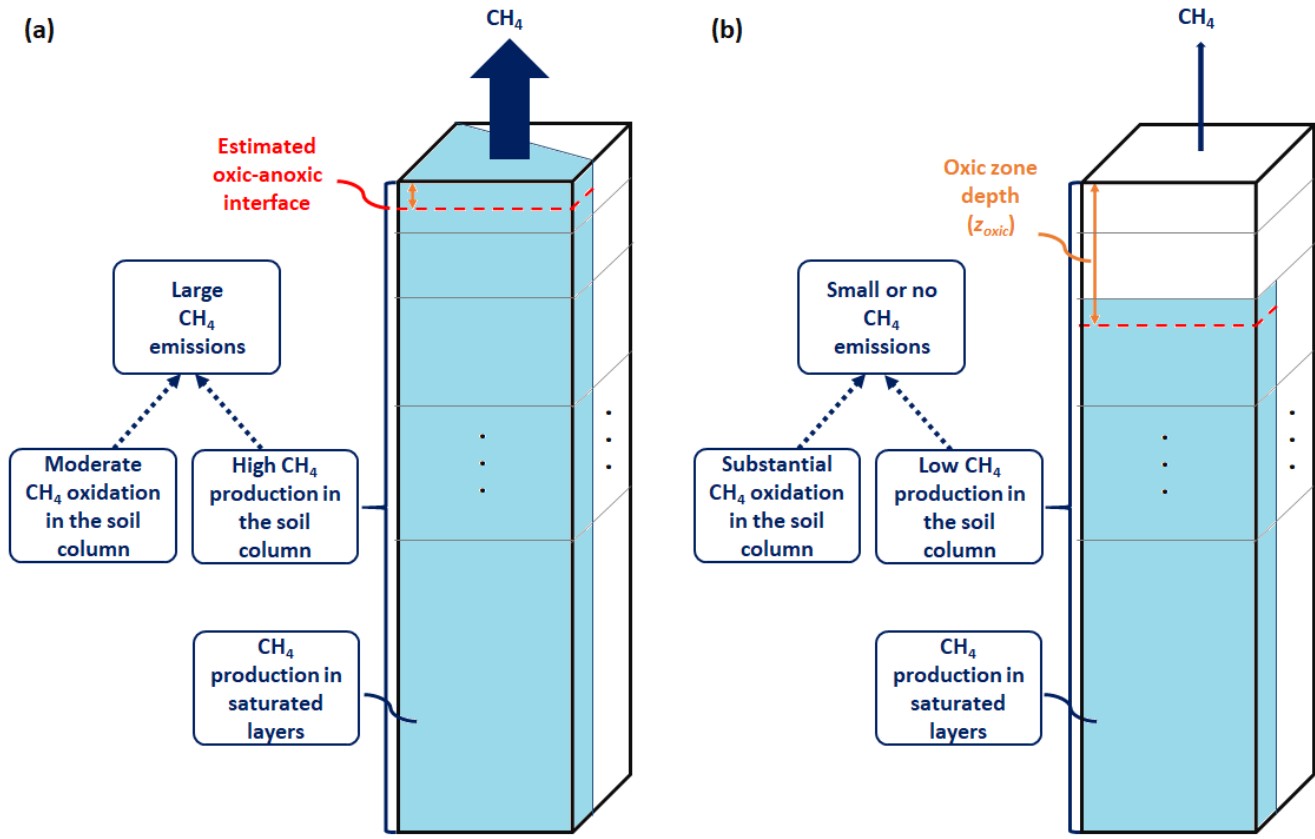


**Figure 2: Illustration of the developed wetland $CH_4$ model (WETMETH) and the dynamics of wetland $CH_4$ processes as represented in the model. This schematic representation depicts a soil column (model grid box) with inundation at the surface (a) and without inundation at the surface (b). The soil column is shown here with multiple layers of unequal thicknesses. The blue area at the surface of (a) represents the inundated surface area. The blue sections in the different soil layers of (a) and (b)**
**represent water-saturated zones. For both (a) and (b), the dashed red horizontal line illustrates the oxic-anoxic interface and the orange vertical arrow shows the relative thickness of the oxic zone or oxic zone depth ($z_{oxic}$). Larger $CH_4$ emissions are expected to occur when the soil surface is flooded than when it is not due to relatively high $CH_4$ production and moderate $CH_4$ oxidation in the soil column.**





### 3.1.1 Parameterization of methane production

For any land location, the rate of $CH_4$ production in an underlying soil layer $i$ ($P_i$ in kg C m$^{-3}$ s$^{-1}$) is parameterized as:

$$P_i = S(\theta_i) \, C_i \, r \, Q_{10}^{\frac{T_i - T_0}{10}} \, \exp\left(-\frac{z_i}{\tau_{\text{prod}}}\right), \tag{1}$$

where $S(\theta_i)$ is the fraction of soil layer that is saturated with water, and $C_i$ is the amount of soil carbon (in kg C m$^{-3}$) in the layer. The product of $S(\theta_i)$ and $C_i$ represents the organic matter (in kg C m$^{-3}$) available for microbial decomposition under anoxic conditions. When the soil surface is not flooded (Fig. 2b), dry soil layers ($S(\theta_i) = 0$) are assumed to be predominantly

oxic and not producing $CH_4$ ($P_i = 0$) mostly due to aerobic microbes dominating the competition for organic matter which results in the starvation of methanogens (Segers, 1998).

The global factor $r$ is the specific $CH_4$ production rate (in kg kg$^{-1}$ s$^{-1}$), which can be defined as the mass of $CH_4$-C that is produced per kilogram of available soil C per unit of time. A meta-analysis of incubated soil samples from various anaerobic landscapes indicates that $r$ can vary between 0.3 to 27.2 µg of $CH_4$-C per g of soil C per day (equivalent to the

range from 3.5 x 10$^{-12}$ to 3.1 x 10$^{-10}$ kg kg$^{-1}$ s$^{-1}$) depending on the landscape type, relative water table position, and soil depth (Treat et al., 2015). Section 4.1 discusses the choice of the value for $r$ as part of the model calibration.

The expression $Q_{10}^{\frac{T_i - T_0}{10}}$, which depends on the average layer temperature $T_i$ (in Kelvin, K) and a baseline temperature $T_0$ (273.15 K), represents the temperature-dependency of $CH_4$ production expressed with a $Q_{10}$ coefficient as commonly done to approximate the sensitivity of biological processes to a temperature change of 10 K (Hegarty, 1973).

While some biological processes double in rate with a warming of 10 K, several studies report a higher temperature sensitivity for $CH_4$ production (i.e. $Q_{10} > 2$) although with large uncertainties (Lupascu et al., 2012; Sjögersten et al., 2018; Walz et al., 2017; Whalen, 2005). Nevertheless, a meta-analysis of temperature-response studies suggests an average $Q_{10}$ of about 4.2 for $CH_4$ production in pure cultures of methanogens (Hoehler and Alperin, 2014; Yvon-Durocher et al., 2014) in agreement with previous estimates (Blodau, 2002). In order to account for uncertainties with this coefficient and define the

occurrence of an optimal temperature for $CH_4$ production (Blake et al., 2015; Dean et al., 2018; Dunfield et al., 1993), a temperature-dependent $Q_{10}$ is considered in WETMETH. Its mathematical formulation is $Q_{10}(T_i) = 1.7 + 2.5 \tanh [0.1 (T_{ref} - T_i)]$, where $T_{ref} = 308.15$ K is a reference temperature (Table 1). This formulation is defined following an expression used for soil respiration in another study (Wu et al., 2016). Additional information on this formulation and its implications for the temperature-dependency of $CH_4$ production are provided in Appendix A1. Furthermore, $CH_4$ production

in WETMETH is assumed to shut down in frozen soil layers although research suggests that slow microbial activity can occur at temperatures below 273.15 K (Panikov and Dedysh, 2000; Rivkina et al., 2004).

The expression $\exp\left(-\frac{z_i}{\tau_{\text{prod}}}\right)$, which depends on the depth of the soil layer $i$ relative to the surface ($z_i$ in m, positive downwards), describes the declining effect of various environmental controls on $CH_4$ production with depth that are generally unresolved by ESMs. These environmental factors include the quality of organic matter and the spread of

methanogens among other factors (Bridgham et al., 2013; Koven et al., 2015; Treat et al., 2015; Walz et al., 2017; Wild et



al., 2016). Here, $\tau_{\mathrm{prod}}$ (in m) is a scaling parameter for CH$_4$ production. The choice of the value for $\tau_{\mathrm{prod}}$ is discussed later as

part of the model calibration (see Section 4.1).

**Table 1: Model parameters for methane production and oxidation.**

| Parameter | Description | Units | Chosen value |
|-----------|-------------|-------|--------------|
| $r$ | Specific CH$_4$ production rate | kg kg$^{-1}$ s$^{-1}$ | [a] 2.6 x 10$^{-10}$ |
| $Q_{10}$ | Temperature coefficient for CH$_4$ production | — | [b] 4.2 |
| $T_{ref}$ | Reference temperature for CH$_4$ production | K | [c] 308.15 |
| $\tau_{\mathrm{prod}}$ | Scaling parameter for CH$_4$ production | m | 0.75 |
| $z_{\mathrm{oatz}}$ | Thickness of the oxic-anoxic transition zone | m | 0.05 |
| $\tau_{\mathrm{oxid}}$ | Scaling parameter for CH$_4$ oxidation | m | 0.0146 |

[a] This value is equivalent to 22.8 μg CH$_4$-C produced per g of soil C per day; [b] A temperature-dependent
$Q_{10}$, approximating 4.2 for a wide range of temperatures, is used instead (see Appendix A1); [c] The reference temperature is
used to define an optimal temperature for CH$_4$ production (see Appendix A1).

The total amount of CH$_4$ produced in the soil column (P in kg C m$^{-2}$ s$^{-1}$) is calculated as:

$$P = \int_{i=1}^{i=k} P_i \, dz_i \, , \qquad (2)$$

where $P_i$ (in kg C m$^{-3}$ s$^{-1}$) is the rate of CH$_4$ production in the soil layer *i* from Eq. (1), dz$_i$ (in m) is the thickness of the soil

layer *i*, and *k* represents the bottom-most soil layer. This amount of CH$_4$ (P) is then subject to oxidation in transit to emission

into the atmosphere.

### 3.1.2 Parameterization of methane oxidation and net methane emissions

Methane oxidation is parameterized based on the amount of CH$_4$ produced in the soil column and the relative thickness of

the oxic zone. Specifically, the total amount of CH$_4$ oxidized in the soil column (O$_x$ in kg C m$^{-2}$ s$^{-1}$) and net CH$_4$ emissions to

the atmosphere (E in kg C m$^{-2}$ s$^{-1}$) are calculated as:

$$O_x = P \, (1 - \exp(-\frac{z_{\mathrm{oxic}}}{\tau_{\mathrm{oxid}}})), \qquad (3)$$

$$E = P - O_x \, , \qquad (4)$$

which is equivalent to the following expression:

$$E = P \, \exp(-\frac{z_{\mathrm{oxic}}}{\tau_{\mathrm{oxid}}}) \, , \qquad (5)$$

where P (in kg C m$^{-2}$ s$^{-1}$) is the total amount of CH$_4$ produced in the soil column as defined in Eq. (2), $z_{\mathrm{oxic}}$ (in m) is the

relative depth (positive downwards) to the oxic-anoxic interface (Fig. 2), and $\tau_{\mathrm{oxid}}$ (in m) is a scaling parameter for CH$_4$

oxidation. As for $\tau_{\mathrm{prod}}$, the choice of the value for $\tau_{\mathrm{oxid}}$ is discussed as part of the model calibration (see Section 4.2).

Regarding $z_{\mathrm{oxic}}$, we assume that O$_2$ may be present in soil layers unsaturated with water as well as in a shallow

oxic-anoxic transition zone within the upper-most soil layer saturated with water (Fig. 2). In this first development of

WETMETH, we consider a constant thickness ($z_{\mathrm{oatz}}$) of 0.05 m for the oxic-anoxic transition zone (Frolking et al., 2002;





Singleton et al., 2018). The penetration of $O_2$ into the soil and its dynamics with changing moisture conditions can be complex depending on site-specific factors such as the soil composition (Estop-Aragonés et al., 2012) and the presence of vascular plants (Brune et al., 2000). In addition, methanotrophs may be present at depth ($> 0.05$ m) below the water table

probably following some adaptation to low $O_2$ conditions (Singleton et al., 2018). Nevertheless, the approach applied here for $z_{oxic}$ is reasonable for ESMs not resolving $O_2$ dynamics and microbial communities in the soil.

For Eq. (3), the expression $(1 - \exp(-\frac{z_{oxic}}{\tau_{oxid}}))$ represents the fraction of P that gets oxidized in transit to emission into the atmosphere. Various studies report estimates of $CH_4$ oxidation as a fraction of produced $CH_4$ in the soil column (Blazewicz et al., 2012; Le Mer and Roger, 2001; Roslev and King, 1996; Segers, 1998; Singleton et al., 2018). From

sample-to-sample and site-to-site, however, $CH_4$ oxidation exhibits a broad range of values ranging from less than 20% to more than 95% depending on the sampled soil depth ranges, whether or not potential $CH_4$ oxidation under anoxic conditions is considered, the monitored transport mechanisms for $CH_4$ among many other factors (Blazewicz et al., 2012; Couwenberg et al., 2010; Jauhiainen et al., 2005; Kwon et al., 2019; Le Mer and Roger, 2001; Moosavi and Crill, 1998; Roslev and King, 1996; Segers, 1998; Singleton et al., 2018; Whalen, 2005). Nevertheless, the largest fractions of oxidized $CH_4$ are generally

associated with the deepest water tables or oxic-anoxic interfaces (Bridgham et al., 2013; Couwenberg et al., 2010; Jauhiainen et al., 2005; Roslev and King, 1996; Segers, 1998; Whalen, 2005).

The parameterization described in Eq. (3) is a simple approach for characterizing $CH_4$ oxidation in the soil column. Such a parameterization is practical when there is little knowledge on the soil chemistry (e.g. $O_2$ and alternate electron acceptors), the dynamics of methanotrophs and other environmental factors exerting a control on $CH_4$ oxidation (Blazewicz

et al., 2012; Blodau, 2002; Dean et al., 2018; Kwon et al., 2019; Singleton et al., 2018; Smemo and Yavitt, 2011). Most importantly, this parameterization considers the net effect of all mechanisms transporting $CH_4$ from the anoxic soil layers where the gas is produced to the atmosphere. The oxidized $CH_4$ is assumed to produce $CO_2$ that becomes part of the soil respiration routinely simulated by ESMs.

## 3.2 The embedding Earth system model

WETMETH has been embedded in the University of Victoria Earth System Climate Model (UVic ESCM), an Earth system model of intermediate complexity (EMIC) (Weaver et al., 2001). A modified version of the EMIC based on UVic ESCM 2.9 (Eby et al., 2009) is used here. The UVic ESCM consists of a 3-D ocean general circulation model that is coupled to a dynamic-thermodynamic sea ice model, a 2-D (vertically-integrated) energy-moisture balance model for the atmosphere and a land surface model (Weaver et al., 2001). The land surface model is a modified version of the Met Office Surface

Exchange Scheme (MOSES) with 14 ground layers of unequal thickness extending down to a depth of 250 m that can simulate permafrost processes such as freeze-thaw dynamics (Avis et al., 2011). The top eight ground layers (~10 m in total depth) are soil layers and contribute to the water cycle, whereas the bottom six ground layers are bedrock layers (Avis et al., 2011). In the hydraulically active layers, porosity and permeability are determined based on the relative abundance of





prescribed sand, clay, and silt-sized particles. Water phase changes are determined over a range of soil temperatures to
determine the fraction of frozen and unfrozen water in the ground (Avis et al., 2011). All components of the UVic ESCM
have a horizontal grid resolution of 3.6° in longitude and 1.8° in latitude (Eby et al., 2009; Weaver et al., 2001).

Wetlands in the UVic ESCM are identified in grid cell areas based on soil moisture content and topography. Model
grid cells in which wetlands can occur are those with unfrozen soil moisture contents greater than 65% of the saturated
moisture content in the upper soil layer for at least one day in a year (Avis et al., 2011). Instead of using a fixed global
threshold value for topography (Avis et al., 2011), the version of the UVic ESCM used here identifies wetland coverage at
the sub-grid scale following a TOPMODEL approach for global models (Gedney and Cox, 2003). Appendix A2 describes a
minor modification applied to this TOPMODEL approach. Section 5.1 presents an evaluation of wetlands simulated by the
UVic ESCM.

The UVic ESCM includes a representation of the global carbon cycle. The terrestrial carbon cycle is simulated
using the Top-down Representation of Interactive Foliage and Flora including Dynamics (TRIFFID), a dynamic global
vegetation model that is coupled to the land surface model (Avis et al., 2011; Meissner et al., 2003). TRIFFID defines the
state of the terrestrial biosphere in terms of soil carbon as well as the structure and coverage of five plant functional types
(PFTs): broadleaf trees, needleleaf trees, shrubs, C3 grasses and C4 grasses (Cox, 2001; Matthews et al., 2004; Meissner et
al., 2003). Terrestrial carbon gain occurs through photosynthesis that is simulated as a function of atmospheric $CO_2$
concentration, shortwave radiation, air temperature, humidity, and soil moisture. Soil carbon gain occurs through litter-fall
and vegetation mortality. The present-day permafrost carbon pool is simulated by the UVic ESCM following a method that
approximates the effect of long-term freeze-thaw cycles on the vertical distribution of carbon in permafrost-affected soils, a
process referred to as cryoturbation (MacDougall and Knutti, 2016). Soil carbon can occur in the top six ground layers
(~3.35 m in total depth). Terrestrial carbon loss occurs through autotrophic respiration by plants and heterotrophic
respiration by soil microbes (Matthews et al., 2004; Meissner et al., 2003). By configuration, permafrost carbon can only be
lost through microbial respiration and this heterotrophic respiration is assumed to shut down in frozen soil layers
(MacDougall et al., 2012; MacDougall and Knutti, 2016).

The marine carbon cycle in the UVic ESCM is represented with organic and inorganic carbon cycle models (Eby et
al., 2009). The organic carbon cycle is based on marine biology simulated with a nutrient-phytoplankton-zooplankton-
detritus (NPZD) ecosystem model (Schmittner et al., 2008). The inorganic carbon cycle model simulates the air-sea
exchange of $CO_2$ and ocean carbonate chemistry following the protocols of the Ocean Carbon-Cycle Model Intercomparison
Project (OCMIP) (Orr, 1999; Weaver et al., 2001). Dissolved inorganic carbon is treated as a passive tracer that is subject to
ocean circulation (Weaver et al., 2001). Carbonate dissolution in ocean sediments is simulated with a model of respiration in
marine sediments (Archer, 1996; Eby et al., 2009).





## 3.3 Model simulations

For this research, three series of model simulations are performed with the UVic ESCM in its standard fully coupled mode and including WETMETH parameterizations:

1. Firstly, the UVic ESCM is spun up for ~5000 years at year 1850 conditions to allow the model to reach an equilibrium climate state representing the pre-industrial period.

2. Secondly, a transient run from 1850 to 2019 is performed in order to evaluate the model performance. This transient run is based on prescribed $CO_2$ concentration and other forcing data from the fifth phase of the Coupled Model Intercomparison Project (CMIP5) (Taylor et al., 2012). The UVic ESCM is driven by historical data from 1850 to 2005 and by Representative Concentration Pathway (RCP) 8.5 data from 2006 to 2019. Supplementary Fig. S1 illustrates how the simulated historical climate conditions compare to observations in terms of global mean surface air temperature.

3. Thirdly, a set of transient runs from 2000 to 2009 is performed to analyze the model sensitivity to poorly constrained parameters. This set of model simulations (sensitivity runs) is performed by perturbing values of poorly constrained parameters associated with wetland $CH_4$ processes.

## 4 Choice of model parameter values

Here, we describe the choice of three WETMETH parameters ($r$ and $\tau_{\mathrm{prod}}$ for $CH_4$ production; $\tau_{\mathrm{oxid}}$ for $CH_4$ oxidation) as part of the model calibration. These model parameters are tuned to observations from northern high-latitude regions due to the scarcity of large-scale datasets from other regions. The model calibration against northern observations is based on the assumption that tuned parameter values will be valid across the globe, which is an important limitation as it will be discussed later. Nonetheless, this approach is deemed reasonable given the present state of data availability. Section 5.1 describes northern wetlands simulated by the UVic ESCM as part of the model validation.

### 4.1 Methane production parameters

Parameters for $CH_4$ production in WETMETH are calibrated against maximum $CH_4$ production rates measured in laboratory incubations of soil samples from several anaerobic environments across northern high-latitude regions (>50°N). These potential $CH_4$ production rates are obtained from a synthesis dataset, which includes information on other environmental variables such as the relative depth of the soil samples (Treat et al., 2015).

To allow a fair model-data comparison, measured $CH_4$ production rates with corresponding soil bulk density from the sites of origin are converted into units of kg C m$^{-3}$ s$^{-1}$ (see Appendix A3). Furthermore, measurements from landscapes identified as uplands and lakes (in the dataset) are excluded from the dataset used in this model calibration. The remaining





measures are potential $CH_4$ production rates in soil samples from landscapes identified (in the dataset) as wetlands,
335   floodplains and lowlands across Alaska.

In order to set values for $r$ and $\tau_{prod}$ from Eq. (1), the depth profile of simulated $CH_4$ production rates across Alaska
for the year 2000 is tuned to that of the measurements. By setting $r$ to 22.8 µg $CH_4$-C produced per g of soil C per C day
(equivalent to 2.6 x $10^{-10}$ kg kg$^{-1}$ s$^{-1}$) and $\tau_{prod}$ to 0.75 m, we obtain a depth profile of simulated $CH_4$ production rates that
compares fairly well to that of potential $CH_4$ production rates from the laboratory incubations (Fig. 3). These default values
340   for $r$ and $\tau_{prod}$ are listed in Table 1. Section 6 presents a sensitivity analysis on these model parameters.

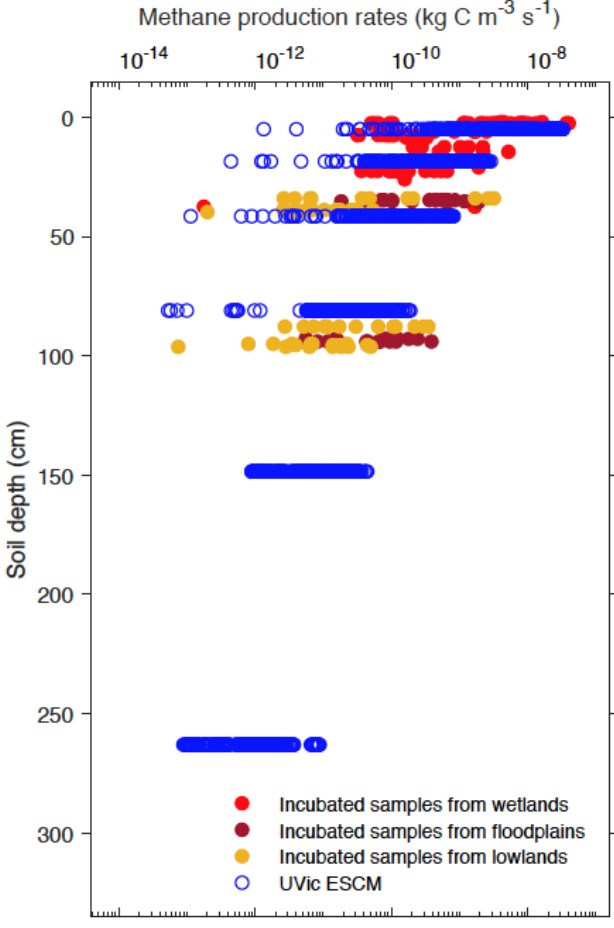

**Figure 3: Vertical profiles of simulated and potential CH₄ production rates from wetlands across Alaska. Potential CH₄
production rates are measurements from laboratory incubations of soil samples collected from various anaerobic ecosystems
(Treat et al., 2015). Both simulated and measured CH₄ production rates are shown here with a log-transformed axis (base-10**
345   **logarithmic scale).**



## 4.2. Methane oxidation parameter

Unlike for $CH_4$ production, there are no published large-scale measurements of $CH_4$ oxidation rates that could be used in this research for the calibration of $CH_4$ oxidation. For that reason, $CH_4$ oxidation in WETMETH is indirectly calibrated via $CH_4$ emissions. A synthesis dataset of seasonal and annual $CH_4$ emissions from various terrestrial sites across temperate, boreal and Arctic regions is used to this end (Treat et al., 2018). The model calibration focuses on annual $CH_4$ emissions from sites north of 50°N for which many data points are available in the dataset.

While most data points are from direct measurements of $CH_4$ emissions, some data points are associated with different modelling methods for estimating $CH_4$ emissions (Treat et al., 2018). To allow a fair model-data comparison, only data points associated with direct measurements of $CH_4$ emissions are included in the model calibration. Furthermore, measurements from lakes, uplands and alpine landscapes are excluded from this model calibration. In particular, the exclusion of data points from uplands and alpine landscapes sorts out measurements of terrestrial $CH_4$ uptake (negative $CH_4$ flux). The retained data points (n = 119) include measurements by chambers (85.7%), flux towers (13.4%) and a combination of flux towers and chambers (0.8%).

The model calibration in this section aims at choosing a value of $\tau_{oxid}$ from Eq. (5) such that the range (minimum - maximum) of annual $CH_4$ emissions across northern wetlands (>50°N) simulated by the UVic ESCM is comparable to that of annual $CH_4$ emissions from the data points (0.1-60.6 g $CH_4$ m$^{-2}$ yr$^{-1}$). By setting $\tau_{oxid}$ to 0.0146 m, we constrain simulated $CH_4$ emissions from northern wetlands (specifically, grid-cell $CH_4$ emissions divided by the inundated fraction of the grid cell) from 2000 to 2009 in the range of 0.04-65.6 g $CH_4$ m$^{-2}$ yr$^{-1}$. This default value for $\tau_{oxid}$ is listed in Table 1. Section 6 presents a sensitivity analysis on this model parameter.

## 5 Evaluation of the model performance

### 5.1 Wetlands

Fig. 4 shows the latitudinal distribution of wetland areas simulated by the UVic ESCM in comparison to two global datasets. The first dataset is Global Inundation Extent from Multi-Satellites (GIEMS), which is based on remotely sensed inundation areas (Papa et al., 2010; Prigent et al., 2001, 2007, 2012). The second dataset is Surface Water Microwave Product Series-Global Lakes and Wetlands Database (SWAMPS-GLWD), which is based on a combination of information from satellites and maps of inundated areas in order to reduce uncertainties associated with the distribution of global wetlands (Poulter et al., 2017). The comparison between the model and the datasets is done over 2000-2007, which is the overlap period for the datasets. Over this period the UVic ESCM simulates an annual maximal extent of ~12.6 million km$^2$ for global wetlands, whereas GIEMS and SWAMPS-GLWD estimate ~9.3 and ~10.6 million km$^2$, respectively.

The UVic ESCM agrees better with SWAMPS-GLWD in regions north of 40°N although with some underestimations around 55°N, and relatively well with GIEMS between 20-40°S (Fig. 4). However, the model simulates too





small wetland areas between 20-30°N when compared to both GIEMS and SWAMPS-GLWD. While our model could be underestimating wetland areas in this latitude zone, inundated areas estimated by GIEMS include rice paddies which prevail in tropical and sub-tropical regions (Prigent et al., 2007, 2012). Rice paddies are likely not represented in SWAMPS-GLWD

as there were efforts to only include natural wetlands during the development of this dataset (Poulter et al., 2017). In comparison to GIEMS and SWAMPS-GLWD, our model simulates small wetland areas in South-East Asia especially near Bangladesh (Fig. 5 and Fig. 6).

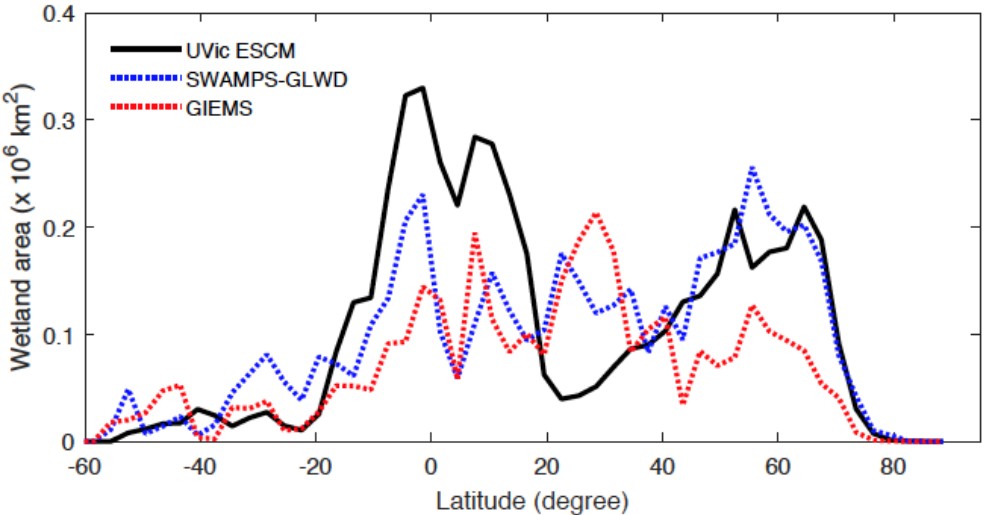

**Figure 4: Latitudinal distribution of wetland areas simulated by the UVic ESCM over the 2000-2007 period in comparison to two**
**global datasets: GIEMS and SWAMPS-GLWD. The comparison period corresponds to the overlap period for the two datasets.**
**The wetland areas are summed across latitude bins of 3°.**

Between 20°N and 20°S, the UVic ESCM simulates a bimodal distribution of the wetland extent that is consistent with the two datasets although the model simulates too large wetland areas (Fig. 4). Unlike for GIEMS and SWAMPS-GLWD, wetlands simulated by the UVic ESCM are widespread in Amazonia, West and Central Africa (Fig. 5 and Fig. 6).

Although the UVic ESCM could be overestimating the extent of wetlands in some of these equatorial regions, it is possible that GIEMS and SWAMPS-GLWD do not detect inundated areas in densely forested regions due to forest canopies. Recent studies suggest that tropical wetlands are commonly underestimated in large-scale datasets (Dargie et al., 2017; Gumbricht et al., 2016).

Conversely, it is possible that the UVic ESCM overestimates tropical wetland areas due to soil hydraulic properties

unrepresented in the model. A potential cause for the overestimation of tropical wetlands in our model is the standard approach for simulating global hydrology in land surface models based on the concentration of only sand, clay and silt in the soil. A recent study suggests that the inclusion of ferralsols (weathered soils with micro-aggregated particles that are common in the humid tropics) in a global terrestrial model can help improve the simulation of tropical wetlands (Gedney et al., 2019).





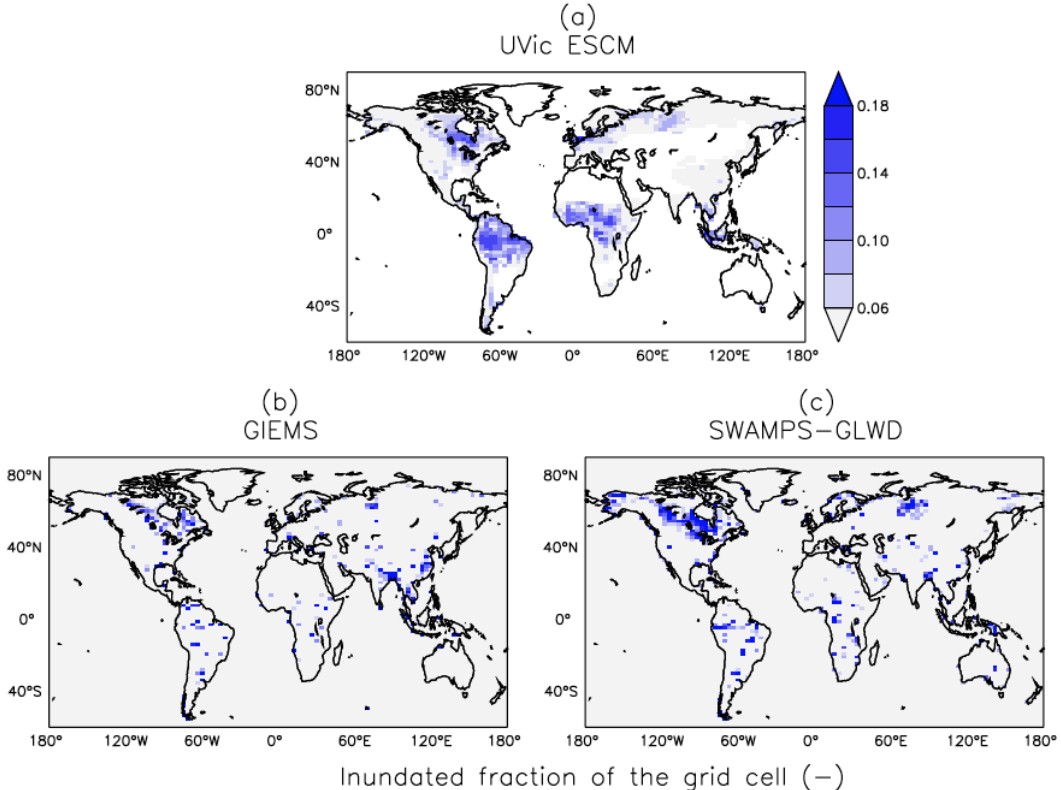


**Figure 5: Average wetland extents (inundated fractions of grid cells) across the globe over the 2000-2007 period as simulated by the UVic ESCM (a) in comparison to two datasets: (b) GIEMS and (c) SWAMPS-GLWD. The datasets are regridded to 3.6° x 1.8° for a fair comparison with the UVic ESCM. The comparison period corresponds to the overlap period for the two datasets.**

Outside of the tropics, the UVic ESCM does a better job at simulating the distribution of wetlands in sub-Arctic and

Arctic regions (Fig. 7). The model simulates the occurrence of wetlands (i.e. surface inundation) across the West Siberian

Lowlands (WSL) in Russia, the Hudson Bay Lowlands (HBL) in Canada as well as over other parts of northern Canada in

agreement with both SWAMPS-GLWD and GIEMS (Fig. 7). However, some disagreements between the UVic ESCM and

the two datasets can also be identified: (i) in comparison to GIEMS, the UVic ESCM simulates more wetland area in the

Hudson Bay Lowlands (HBL) as well as widespread wetlands in parts of northern Eurasia (Fig. 7b and Fig. S2b); (ii) in

comparison to SWAMPS-GLWD, the model simulates less wetland area over the WSL and northern Canada including the

HBL and more wetland area in parts of Europe (Fig. 7c and Fig. S2c).

Statistical analyses show that: (i) the UVic ESCM agrees better with SWAMPS-GLWD than with GIEMS at both

the regional and global scale; and (ii) the model compares better with the two datasets across northern regions than at the

global scale. For details on the statistical evaluation, please see supplementary Table S1.

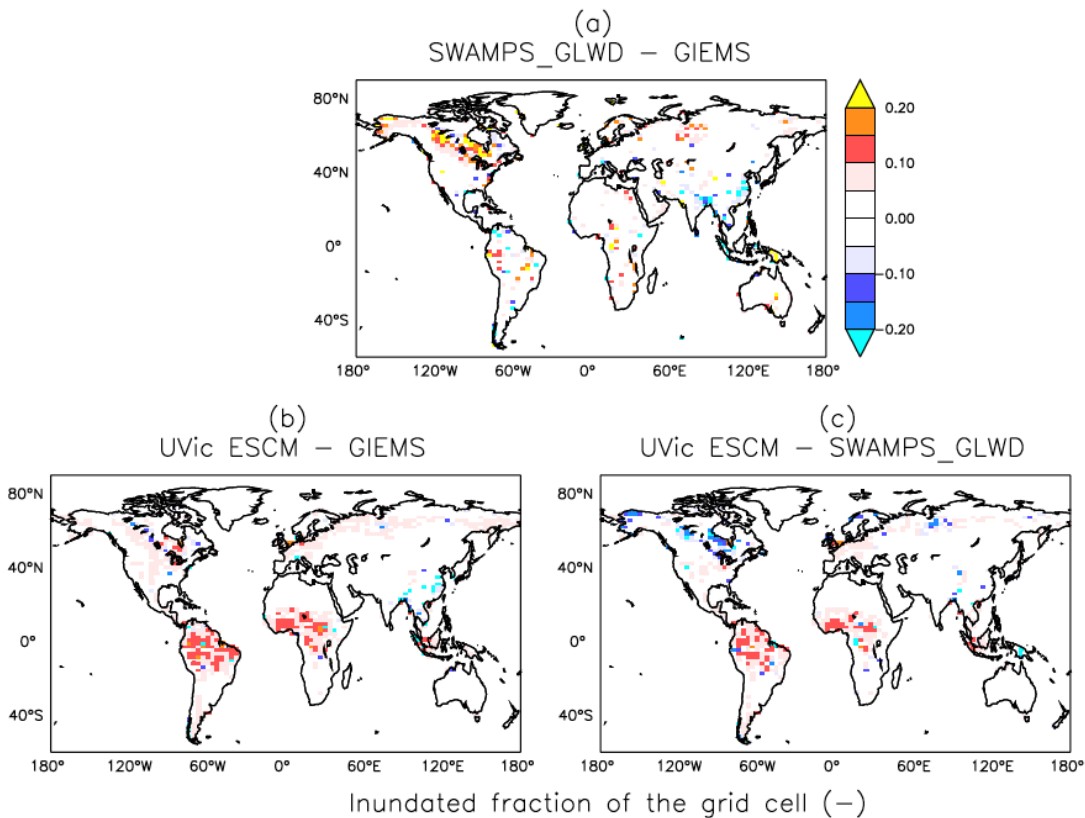


**Figure 6: Differences in global wetland extents (inundated fractions of grid cells) between two datasets (GIEMS and SWAMPS-GLWD) and the UVic ESCM over the 2000-2007 period: (a) SWAMPS-GLWD – GIEMS, (b) UVic ESCM – GIEMS, and (c) UVic ESCM – SWAMPS-GLWD. The comparison period corresponds to the overlap period for the two datasets.**

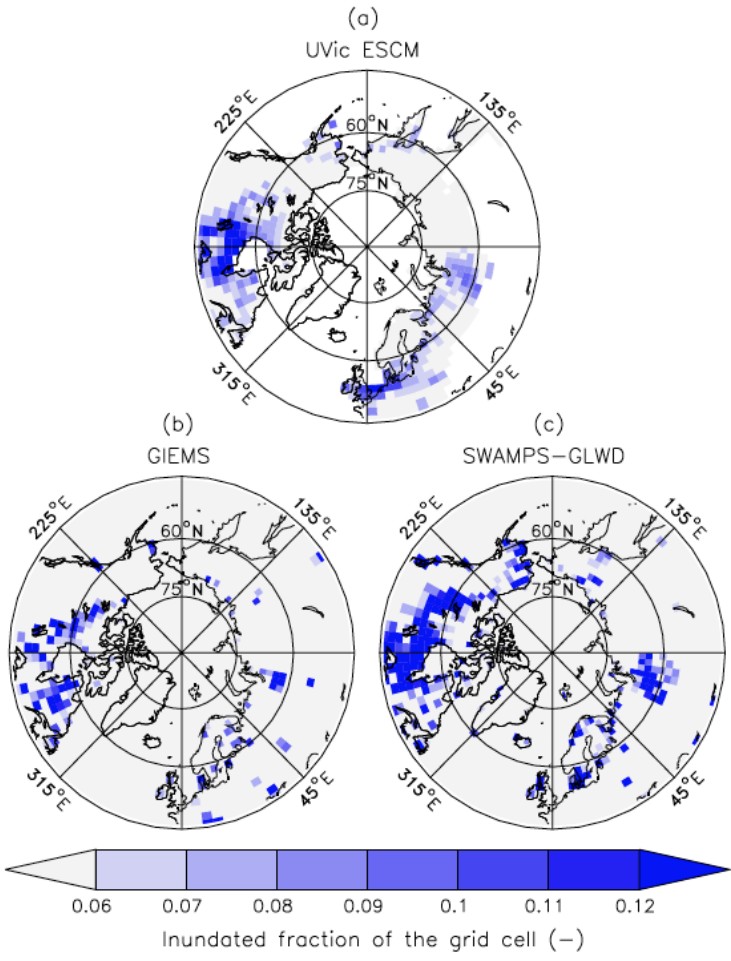

Figure 7: Average wetland extents (inundated fractions of grid cells) in the north of 45°N over the 2000-2007 period as simulated
by the UVic ESCM (a) in comparison to two datasets: (b) GIEMS and (c) SWAMPS-GLWD. The datasets are regridded to 3.6° x
1.8° for a fair comparison with the UVic ESCM. The comparison period corresponds to the overlap period for the two datasets.

**5.2 Wetland methane emissions**

Given the relative coarse grid resolution of the UVic ESCM, the model validation with respect to wetland $CH_4$ emissions

focuses on large-scale emissions such as regional, zonal, and global emissions. Moreover, this model validation focuses on

northern high-latitude regions because observations and estimates of wetland $CH_4$ emissions from other regions (e.g. the

tropics) are scarce. This focus is further justified by the fact that our model better simulates the distribution of wetlands in

northern high-latitude regions than in the tropics (see Section 5.1). Indeed, the extent of wetlands is a major control for

wetland $CH_4$ emissions simulated by process-based models and probably the primary contributor to related uncertainties

(Melton et al., 2013; Saunois et al., 2020; Zhang et al., 2017a).



### 5.2.1 Northern high-latitude emissions

The UVic ESCM simulates total $CH_4$ emissions from northern wetlands that are in the range of recent estimates. Over the 2013-2014 period, the model simulates mean annual emissions of 33.2 Tg $CH_4$ $yr^{-1}$ for wetlands north of 45°N (Table 2). These $CH_4$ emissions are consistent with estimates from recent upscaled flux measurements (UFMs) over the same period based on a random forest (RF) algorithm and three wetland maps (Peltola et al., 2019): 30.6 ± 9.2 Tg $CH_4$ $yr^{-1}$ (RF-DYPTOP), 31.7 ± 9.4 Tg $CH_4$ $yr^{-1}$ (RF-PEATMAP), and 37.6 ± 11.8 Tg $CH_4$ $yr^{-1}$ (RF-GLWD) (Table 2). Supplementary Table S2 shows that the UVic ESCM has no preferential agreement with one of the three UFMs.

**Table 2: Mean annual wetland $CH_4$ emissions simulated by the UVic ESCM in comparison to estimated emissions from the literature. All emissions are reported in Tg $CH_4$ $yr^{-1}$ and uncertainties are provided for estimates from the literature. Three periods are used to allow a fair comparison between the UVic ESCM and estimates from the literature where possible: 2008-2017 as in the latest global $CH_4$ budget report (Saunois et al., 2020), 2013-2014 as for recent upscaled flux measurements across the northern high-latitudes (Peltola et al., 2019), and 1993-2004 as for the WETCHIMP model ensemble (Melton et al., 2013). Principal methods used in the different references for estimates are reported in the last column: Top-down (TD) methods including inverse models (IM), and bottom-up (BU) methods including upscaled measurements (UM) as well as process-based models (PM).**

| | Geographical delimitation | UVic ESCM period | UVic ESCM emissions | Estimated emissions | Reference for estimates | Method in reference |
|---|---|---|---|---|---|---|
| Hudson Bay Lowlands | 50 – 60°N; 75 – 96°W | 2013-2014 | 2.9 | 2.3 ± 0.3 | Pickett-Heaps et al., 2011 | BU |
| | | | | 2.4 ± 0.3 | Miller et al., 2014 | IM |
| | | | | 2.7 - 3.4 | Thompson et al., 2017 | IM |
| West Siberian Lowlands | 50 – 75°N; 60 – 95 °E | 2013-2014 | 4.1 | 3.9 ± 1.3 | Glagolev et al., 2011 | UM |
| | | | | 6.1 ± 1.2 | Bohn et al., 2015 [a] | IM |
| | | | | 6.9 ± 3.6 | Thompson et al., 2017 | IM |
| Pan-Arctic Wetlands | 60°N – 90°N | 2008-2017 | 17.3 | 7 – 16 | Saunois et al., 2020 | TD |
| | | | | 2 – 18 | Saunois et al., 2020 | BU |
| Northern Wetlands | 40°N – 90°N | 2008-2017 | 38.5 | 37.4 ± 7.2 | Treat et al., 2018 | BU |
| | 45°N – 90°N | 2013-2014 | 33.2 | 30.6 ± 9.2 | Peltola et al., 2019 | UM |
| | | | | 31.7 ± 9.4 | Peltola et al., 2019 | UM |
| | | | | 37.6 ± 11.8 | Peltola et al., 2019 | UM |
| Tropical Wetlands | 30°S – 30°N | 1993-2004 | 105.5 | 126 ± 31 | Melton et al., 2013 [a] | PM |
| | | | | 90 ± 77 | Sjögersten et al. 2014 | UM |
| Global Wetlands | 90°S – 90°N | 2008-2017 | 158.6 | 155 – 200 | Saunois et al., 2020 | TD |
| | | | | 102 – 182 | Saunois et al., 2020 | BU |

[a] These reported estimates are model ensemble means. For the West Siberian Lowlands, the range between the inverse models is 3.1–9.8 Tg $CH_4$ $yr^{-1}$ (Bohn et al., 2015). For tropical wetlands, the range between the process-based models is 85–184 Tg $CH_4$ $yr^{-1}$ (Melton et al., 2013).

Fig. 8 shows the spatial distribution of simulated $CH_4$ emissions in comparison to the three UFMs. When compared to each other, the three UFMs exhibit substantial differences primarily attributed to the distinct wetland distributions (Peltola et al., 2019). Considering the general pattern and magnitude of wetland $CH_4$ emissions, the UVic ESCM agrees with either




two or all three UFMs over key source regions such as the Hudson Bay Lowlands (HBL), the West Siberian Lowlands (WSL), western Europe and south-central Canada (Fig. 8).

The UVic ESCM simulates less $CH_4$ emissions over parts of northeastern Canada and Fennoscandia in comparison to the UFMs (Fig. 8). However, the three UFMs do not necessarily agree on both the distribution and magnitude of wetland $CH_4$ emissions in these regions. Furthermore, the UVic ESCM does not simulate wetland $CH_4$ emissions in southern Eurasia (40-135°E; 45-60°N) while the three UFMs suggest that $CH_4$ can be emitted from sporadic wetlands in this region (Fig. 8). Overall, the mismatch between the UFMs and our model in terms of northern $CH_4$ emissions can be primarily attributed to

differences in the wetland extent, but also to the spatial distribution of soil carbon simulated by the UVic ESCM (MacDougall and Knutti, 2016).

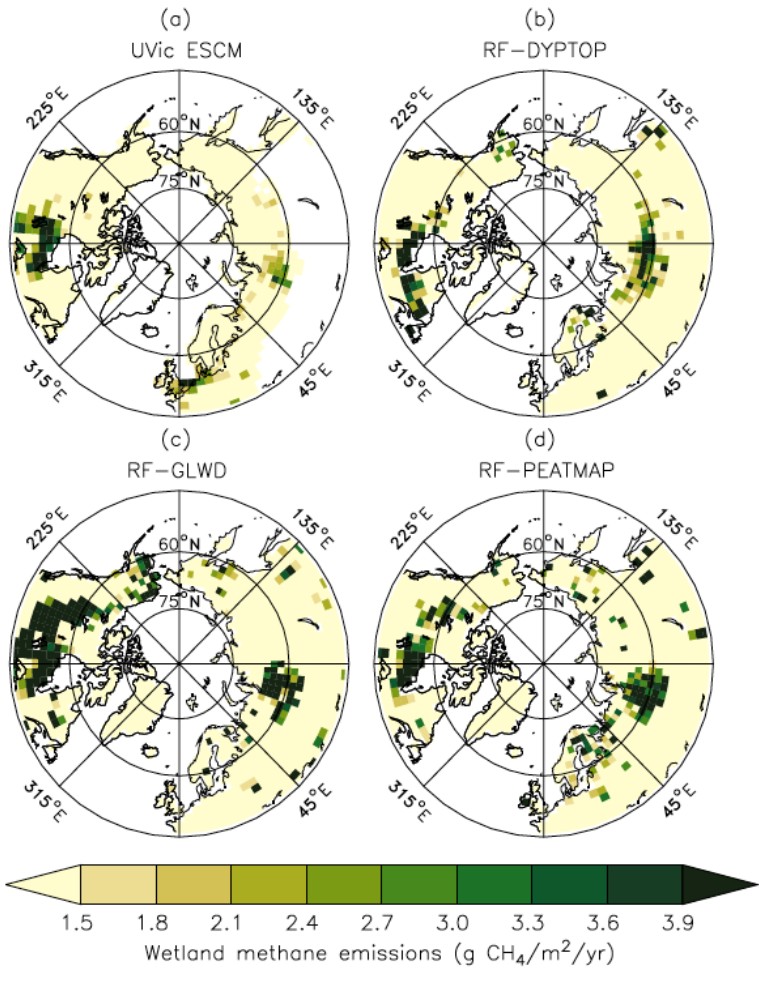

**Figure 8: Average $CH_4$ emissions from wetlands north of 45°N over the 2013-2014 period as simulated by the UVic ESCM (a) in comparison to three datasets (upscaled flux measurements): (b) RF-DYPTOP, (c) RF-GLWD and (d) RF-PEATMAP. The**
**datasets are regridded to 3.6° x 1.8° for a fair comparison with the UVic ESCM. The comparison period corresponds to the overlap period for the three datasets.**



In terms of mean annual emissions from key source regions, the UVic ESCM simulates 2.9 Tg $CH_4$ $yr^{-1}$ for the Hudson Bay Lowlands (HBL) over the 2013-2014 period (Table 2). Although these emissions are lower than estimates by the three UFMs (3.1-6.5 Tg $CH_4$ $yr^{-1}$) (Peltola et al., 2019), estimates by inverse models (2.0-3.4 Tg $CH_4$ $yr^{-1}$) over this

region are comparable to our model results (Miller et al., 2014; Pickett-Heaps et al., 2011; Thompson et al., 2017). Furthermore, the UVic ESCM simulates total wetland emissions of 4.1 Tg $CH_4$ $yr^{-1}$ for the West Siberian Lowlands (WSL) over the 2013-2014 period (Table 2). Regional estimates based on the three UFMs are higher (4.9-8.5 Tg $CH_4$ $yr^{-1}$) than our model results over the same period (Peltola et al., 2019), whereas previous observation-based estimates for the WSL suggest regional wetland emissions (3.9 ± 1.3 Tg $CH_4$ $yr^{-1}$) that are similar to our model results (Glagolev et al., 2011). Estimates by

inverse models over the WSL are relatively high but comparable to our model estimates (Table 2): 6.1 ± 1.2 Tg $CH_4$ $yr^{-1}$ (Bohn et al., 2015) and 6.9 ± 3.6 Tg $CH_4$ $yr^{-1}$ (Thompson et al., 2017).

The UVic ESCM is also evaluated with respect to wetland $CH_4$ emissions over the 2000-2009 and 2008-2017 decades, which both are reference periods for the latest global $CH_4$ budget report (Saunois et al., 2020). For wetlands north of 40°N, the UVic ESCM simulates emissions of 37.7 Tg $CH_4$ $yr^{-1}$ over the 2000-2009 decade and 38.5 Tg $CH_4$ $yr^{-1}$ over the

2008-2017 decade. These wetland $CH_4$ emissions are consistent with recent estimates (37.4 ± 7.2 Tg $CH_4$ $yr^{-1}$) from data-constrained model ensembles over the same region (Treat et al., 2018). For wetlands north of 45°N, the model simulates total emissions that are in the range of estimates for the 2013-2014 period discussed earlier (32.4 Tg $CH_4$ $yr^{-1}$ over 2000-2009 and 33.1 Tg $CH_4$ $yr^{-1}$ over 2008-2017). For Pan-Arctic wetlands (>60°N), the UVic ESCM simulates emissions of 17.4 Tg $CH_4$ $yr^{-1}$ over the 2000-2009 decade and a similar amount over the 2008-2017 decade (Table 2). These wetland $CH_4$ emissions

correspond to the upper limit of bottom-up estimates (2-18 Tg $CH_4$ $yr^{-1}$) from the latest global $CH_4$ budget report (Saunois et al., 2020).

Fig. 9 shows seasonal cycles of $CH_4$ emissions from wetlands north of 45°N over the 2013-2014 period as simulated by the UVic ESCM and estimated from the three UFMs (Peltola et al., 2019). The pattern and magnitude of simulated seasonal emissions compare well to that of the UFMs. For both the model and UFMs, minimal emissions vary

between 0.2-0.6 Tg $CH_4$ $month^{-1}$ and occur in December while peak emissions are well below 10 Tg $CH_4$ $month^{-1}$ and occur in July (Fig. 9). However, simulated peak emissions (~8.5 Tg $CH_4$ $month^{-1}$) are relatively higher than peak emissions for the UFMs (range of best estimates: 5.6-7.5 Tg $CH_4$ $month^{-1}$). Moreover, in comparison to the three UFMs, the UVic ESCM simulates lower $CH_4$ emissions between December and May but higher $CH_4$ emissions between July and September (Fig. 9).

The UVic ESCM simulates the occurrence of wetland $CH_4$ emissions during the non-growing season. For wetlands

north of 45°N, our model simulates total emissions of 2.1 Tg $CH_4$ $yr^{-1}$ between November and March. The UFMs predict total emissions of 4.6-10.2 Tg $CH_4$ $yr^{-1}$ during these cold months (Peltola et al., 2019). For wetlands north of 60°N, the UVic ESCM simulates emissions of 1.2 Tg $CH_4$ $yr^{-1}$ from October through May in agreement with recent estimates (1.6 ± 0.1 Tg $CH_4$ $yr^{-1}$) from data-constrained model ensembles for these months (Treat et al., 2018). Based on our calculations, the three UFMs predict about 3.5-4.5 Tg $CH_4$ $yr^{-1}$ emitted from wetlands north of 60°N between October and May. Overall, this





analysis shows that WETMETH is capable of simulating non-negligible $CH_4$ emissions from northern wetlands during cold

months as emphasized by recent studies (Treat et al., 2018; Zona et al., 2016).

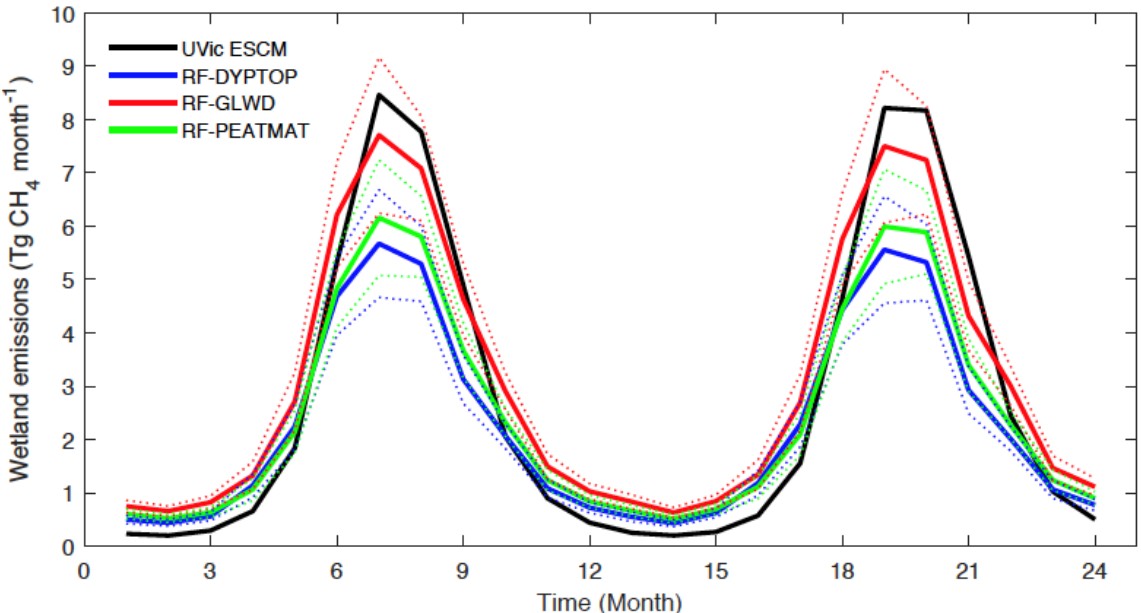

**Figure 9: Seasonal variations of $CH_4$ emissions from wetlands north of 45°N over the 2013-2014 period as simulated by the UVic ESCM in comparison to three upscaled flux measurements (RF-DYPTOP, RF-GLWD and RF-PEATMAP). The dashed lines**
**show the uncertainty range for the upscaled flux measurements.**

### 5.2.2 Global emissions

The UVic ESCM simulates total emissions of 155.1 and 158.6 Tg $CH_4$ $yr^{-1}$ from global wetlands over the 2000-2009 and

2008-2017 decades, respectively. According to the latest global $CH_4$ budget report, these wetland emissions are in the mid-

range of bottom-up estimates (102-179 and 102-182 Tg $CH_4$ $yr^{-1}$) but close to the lower limit of top-down estimates (153-

196 and 155-200 Tg $CH_4$ $yr^{-1}$) over the two decades (Saunois et al., 2020). Previous bottom-up estimates are significantly

high (Melton et al., 2013; Saunois et al., 2016) primarily due to possible double counting of emissions from wetlands and

other inland water areas (Saunois et al., 2020; Thornton et al., 2016) in addition to uncertainties associated with the extent of

wetlands and model parameterizations (Melton et al., 2013). Table 2 summarizes the comparison between the model results

and estimates from the latest global $CH_4$ budget report for the 2008-2017 decade.

Fig. 10 shows the spatial distribution of simulated wetland $CH_4$ emissions over the 2001-2004 period in comparison

to three process-based model ensembles: GCP-CH4 (Poulter et al., 2017), WetCHARTs (Bloom et al., 2017), and

WETCHIMP (Melton et al., 2013). The UVic ESCM simulates few $CH_4$-emitting areas over South-East Asia in comparison

to the three model ensembles. The potential underestimation of wetland $CH_4$ emissions in that region is associated with the

relatively few wetland areas simulated by the UVic ESCM (see Section 5.1). In tropical Africa, our model simulates too

many $CH_4$-emitting locations in comparison to the model ensembles (Fig. 10), which is also associated with the distribution



of simulated wetlands (see Section 5.1). Nevertheless, the UVic ESCM simulates the occurrence of wetland $CH_4$ emissions in key source regions such as the Amazon and Congo River basins, South Sudan (Sudd swamps), and Indonesian islands (Fig. 10). For the Amazon and Congo River basins, however, the UVic ESCM simulates lower wetland $CH_4$ emissions than predicted by the model ensembles (Fig. 10). This can be due to either the consideration of an optimal temperature for $CH_4$

production (around 27°C) in our model unlike many other process-based models, or the fact that model parameters in this study are tuned to northern estimates.

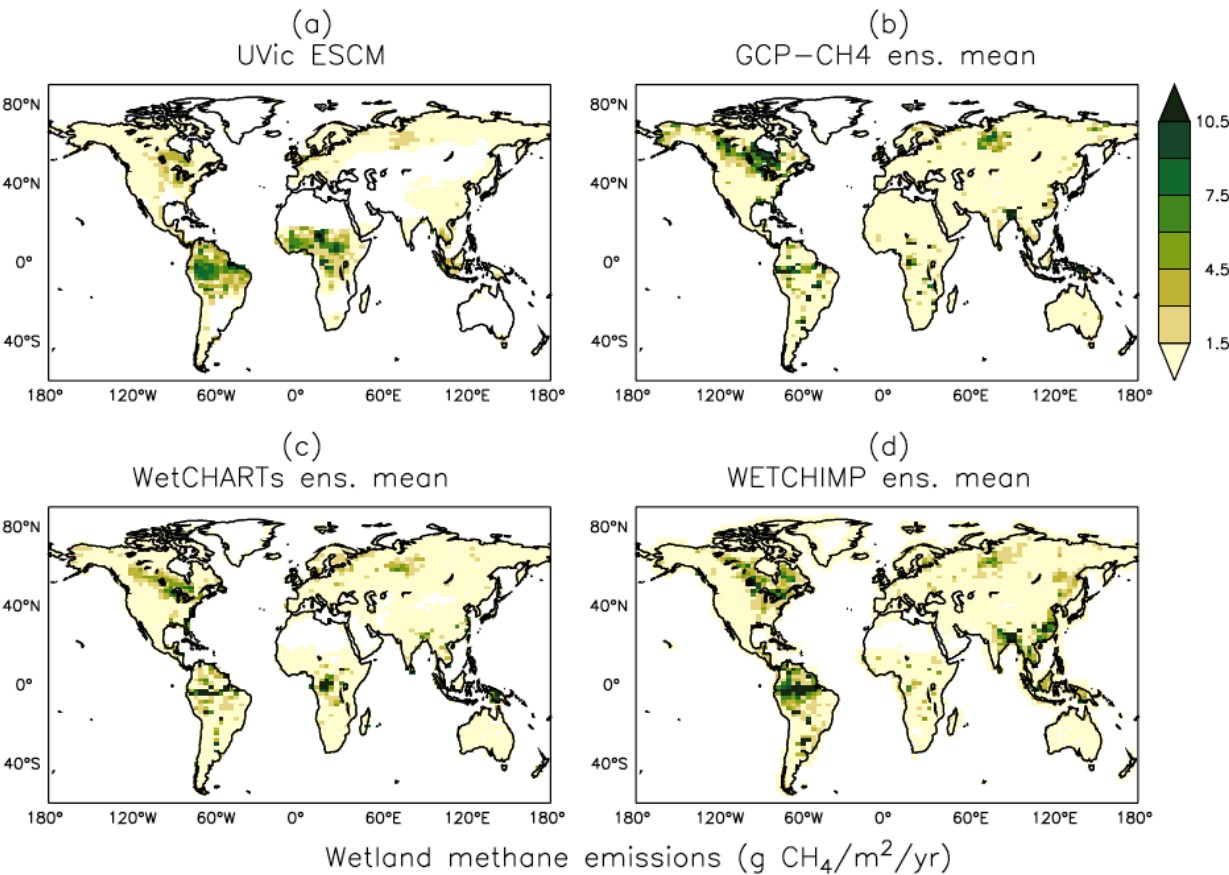

**Figure 10: Average methane emissions from global wetlands over the 2001-2004 period as simulated by the UVic ESCM (a) in comparison to three process-based model ensembles: (b) GCP-CH4, (c) WetCHARTs, and (d) WETCHIMP. The model ensembles**
**are regridded to 3.6° x 1.8° for a fair comparison with the UVic ESCM. The comparison period corresponds to the overlap period for the three model ensembles.**

Fig. 11a shows the latitudinal distribution of simulated wetland $CH_4$ emissions in comparison to the model ensembles. Interestingly, although GCP-CH4 and WetCHARTs are based on the same wetland dataset (SWAMPS-GLWD) (Bloom et al., 2017; Poulter et al., 2017), their zonal wetland $CH_4$ emissions are very different especially near the Equator

and across northern high-latitude regions (Fig. 11a).





Using the three model ensembles as reference, the UVic ESCM simulates significantly lower wetland $CH_4$ emissions around the Equator (Fig. 11a), despite that the model simulates too large equatorial wetland areas (Fig. 4). In fact, wetland emission intensities (emissions per unit of wetland area) by the UVic ESCM are lower than those by the model ensembles between 10°S and 10°N (Fig. 11b) due to relatively large wetland areas but small $CH_4$ emissions in equatorial

regions (Fig. 4 versus Fig. 11a). As previously discussed, the relatively small $CH_4$ emissions simulated by the UVic ESCM in equatorial regions can be associated with either the optimal temperature for $CH_4$ production considered in WETMETH but not in most other process-based models, or the fact that model parameters in this study are tuned to northern estimates.

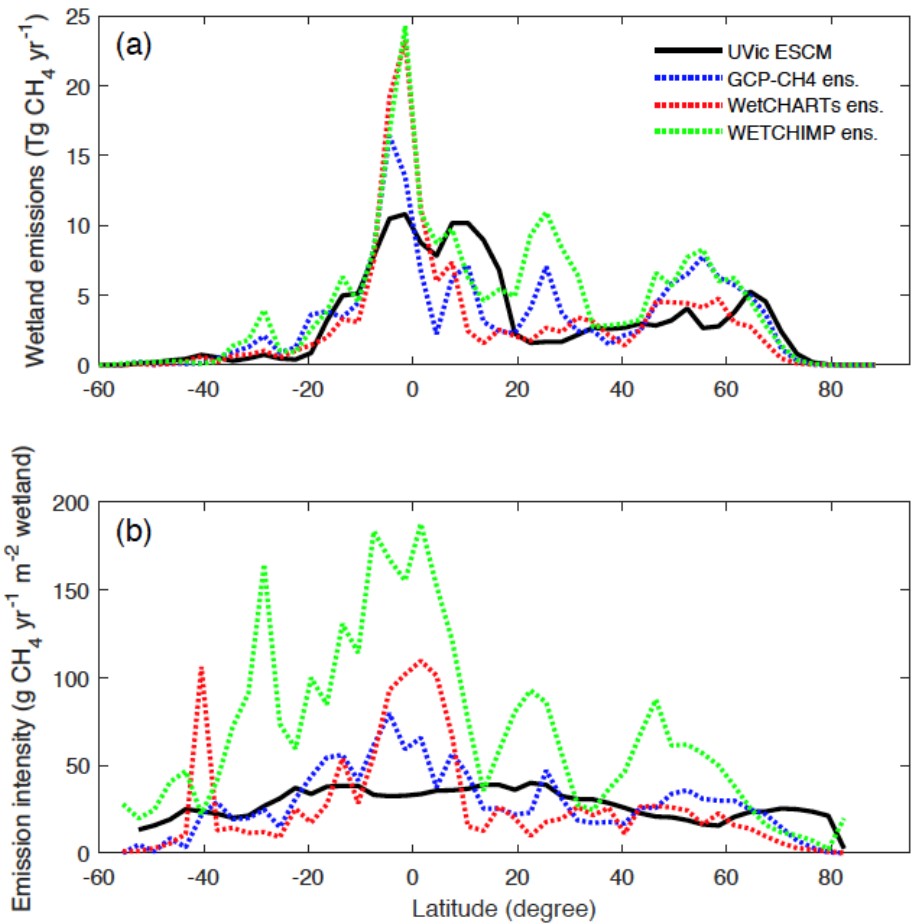

**Figure 11: (a) Latitudinal distribution of wetland methane emissions simulated by the UVic ESCM over the 2001-2004 period in**
**comparison to three process-based model ensembles: GCP-CH4, WetCHARTs and WETCHIMP. The comparison period corresponds to the overlap period for the three model ensembles. (b) Latitudinal emission intensity (methane emissions per unit of wetland area) simulated by the UVic ESCM over the 2001-2004 period in comparison to the three process-based model ensembles. GCP-CH4 and WetCHARTs both use SWAMPS-GLWD as prescribed wetlands. The wetland methane emissions and emission intensities are summed across latitude bins of 3°.**

Furthermore, the UVic ESCM simulates more wetland $CH_4$ emissions between 10-20°N than the three model ensembles (Fig. 11a) and this can be attributed to the widespread wetlands in West and Central Africa simulated by our





model (Fig. 5 and Fig. 6). In addition, the UVic ESCM simulates significantly less wetland $CH_4$ emissions between 20-35°N in comparison to the WETCHIMP ensemble (Fig. 11a) and this can be attributed to the relatively small wetland areas simulated by the UVic ESCM in South-East Asia where some models include agricultural wetlands such as rice paddies.

Moreover, wetland emission intensities by the UVic ESCM feature low variability with latitude unlike the three model ensembles (Fig. 11b). Such a relative lack of variability can be attributed to two factors: (i) both wetland areas and $CH_4$ emissions simulated by the UVic ESCM feature relatively low variability with latitude compared to the datasets and model ensembles (Fig. 4 and Fig. 11a); and (ii) as previously discussed, our model likely simulates too large wetland areas but too small $CH_4$ emissions around the Equator implying a lack of variability across tropical latitudes.

Despite the various discrepancies between the UVic ESCM and both model ensembles regarding the distribution of wetland $CH_4$ emissions in the tropics, our model simulates mean annual $CH_4$ emissions from tropical wetlands that are in the range of estimates from the literature (Table 2). For the 1993-2004 period, the UVic ESCM simulates tropical wetland $CH_4$ emissions of 105.5 Tg $CH_4$ yr$^{-1}$ whereas the WETCHIMP ensemble predicts 126 ± 31 Tg $CH_4$ yr$^{-1}$ (Melton et al., 2013). Another study suggests a lower mean value (90 ± 77 Tg $CH_4$ yr$^{-1}$) for wetland $CH_4$ emissions in the tropics although with

large uncertainties (Sjögersten et al., 2014). Indeed, several studies indicate that wetland $CH_4$ emissions in the tropics are highly uncertain due to limited ground-based measurements and poorly delimitated wetland extent (Dargie et al., 2017; Gumbricht et al., 2016; Hu et al., 2018; Pangala et al., 2017; Saunois et al., 2020).

## 6 Model sensitivity to poorly constrained parameters

We performed a set of 30 model runs with perturbed parameter values (sensitivity runs) over the 2000-2009 decade in order

to analyze the model sensitivity to poorly constrained parameters ($T_{ref}$, $r$, $\tau_{prod}$, $z_{oatz}$, and $\tau_{oxid}$). For each parameter, we increased or decreased the default value by 10, 20, and 30% while holding constant values for other parameters (fixed to default values). We then compared results from the sensitivity runs to the model simulation with all parameter values set to default values (control run). This comparison focuses on the total simulated global (90°S-90°N), northern (45-90°N), and tropical (30°S-30°N) wetland $CH_4$ emissions over the 2000-2009 decade.

Our results show that the model sensitivity varies with the different parameters and across regions (Fig. 12). Among the five poorly constrained parameters, the UVic ESCM is most sensitive to perturbations of the two parameters for $CH_4$ oxidation ($z_{oatz}$ and $\tau_{oxid}$) at both the global and regional scale. For $z_{oatz}$, a decrease (increase) of the default parameter value by 10-30% results in an augmentation (reduction) of default wetland $CH_4$ emissions by 41-179% (29-64%) at both the global and regional scale (Fig. 12j-l). For $\tau_{oxid}$, a decrease (increase) of the default parameter value by 10-30% implies a

reduction (augmentation) of default wetland $CH_4$ emissions by 32-77% (37-120%) at both the global and regional scale (Fig. 12m-o).

The UVic ESCM is also very sensitive to perturbations of $T_{ref}$, but this sensitivity is more pronounced for tropical regions than northern regions (Fig. 12a-c). For northern regions, a decrease (increase) of $T_{ref}$ by 10-30% results in a





reduction (augmentation) of default wetland CH₄ emissions by 5-21% (3-5%). For tropical regions, however, a decrease

(increase) of $T_{ref}$ by 10-30% results in a reduction (augmentation) of default wetland CH₄ emissions by 34-82% (33-75%).

Globally, a decrease (increase) of $T_{ref}$ by 10-30% results in a reduction (augmentation) of default wetland CH₄ emissions by

26-66% (24-55%). The model sensitivity to perturbations of $r$ is linear across all regions (Fig. 12d-f). Lastly, the model is

least sensitive to perturbation of $\tau_{prod}$ across the globe (Fig. 12g-i).

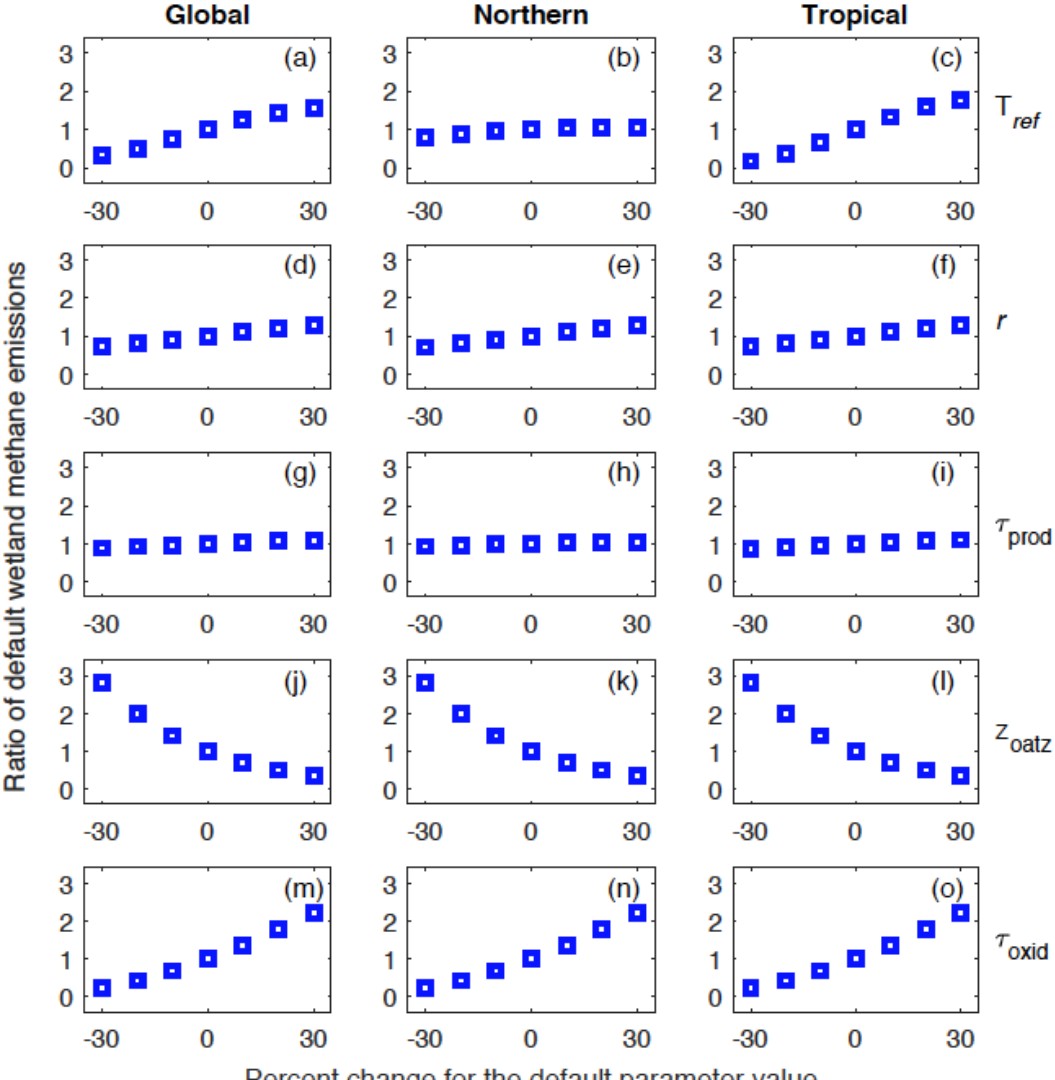

**Figure 12: Analysis of the model sensitivity to perturbations of poorly constrained parameters: $T_{ref}$, $r$, $\tau_{prod}$, $z_{oatz}$, and $\tau_{oxid}$. For each parameter, the default value is increased or decreased by 10, 20, and 30% while values of other parameters are held constant (to default values). The model sensitivity is analyzed with respect to global (90°S-90°N), northern (45-90°N), and tropical (30°S-30°N) wetland methane emissions. Vertical axes show the ratio of the resulting emissions to the default emissions.**





# 7 Discussions

## 7.1 WETMETH in the spectrum of wetland methane models

A recent study reviewed 40 models of $CH_4$ emissions in terrestrial ecosystems (predominantly rice paddies and natural wetlands) and classified them into three categories based on their level of complexity: relatively simple models, relatively mechanistic models, and mechanistic models (Xu et al., 2016). Relatively simple models are those that simulate net $CH_4$ emissions based on soil carbon or other environmental factors without explicit representations for the different $CH_4$

production and oxidation pathways as well as mechanisms transporting $CH_4$ to the atmosphere. Relatively mechanistic models are those that account for at least one transport mechanism for $CH_4$ release in addition to representing $CH_4$ production and oxidation with simple functions. Mechanistic models are more comprehensive and explicitly simulate different pathways for both $CH_4$ production and oxidation, more than two mechanisms for $CH_4$ release, as well as their environmental controls. Based on this classification, WETMETH is a relatively simple model in the sense that it does not

distinguish pathways for $CH_4$ production and oxidation as well as the various mechanisms transporting $CH_4$ to the atmosphere.

Although some wetland $CH_4$ models are claimed to be embedded in ESMs (Xu et al., 2016), none of these models are currently run in fully coupled models with feedbacks between climate conditions and the global carbon cycle. Most of these models are rather implemented in dynamic vegetation models or uncoupled land surface components of climate models

(Arora et al., 2018; Eliseev et al., 2008; Hodson et al., 2011; Riley et al., 2011; Ringeval et al., 2011; Wania et al., 2009). Nonetheless, relatively simple models present the ideal level of complexity for the current generation of ESMs. More complex models generally imply detailed soil chemistry for $O_2$ and alternate electron acceptors (Riley et al., 2011; Wania et al., 2010), different carbon substrates and their effects on $CH_4$ production (Grant, 1998; Lovley and Klug, 1986), an explicit representation of the dynamics of different microbial communities (Grant, 1998; Xu et al., 2015), which all require

comprehensive soil chemistry or model parameters that are currently not common in ESMs (Xu et al., 2016). Process parameterizations in mechanistic models generally imply too many degrees of freedom, making it difficult to constrain model parameters against sparse observations. Furthermore, mechanistic models may be too demanding computationally for fully coupled ESM runs without a proportional benefit for large-scale simulations of wetland $CH_4$ emissions.

The particularity of WETMETH among relatively simple models is that the model accounts for an optimum

temperature for $CH_4$ production, a depth-dependent representation for $CH_4$ production allowing a calibration of parameters against potential $CH_4$ production rates from laboratory incubations, dynamic $CH_4$ oxidation based on the vertical distribution of soil moisture, and the potential for $CH_4$ emissions in non-inundated ecosystems with relatively high level of soil moisture content. In conclusion, WETMETH is simple enough to be compatible with ESMs and yet complex enough to simulate in an implicit way biogeochemical processes regulating wetland $CH_4$ emissions.





## 7.2 Limitations for WETMETH

The developed wetland $CH_4$ model is associated with several limitations, which are linked to either its level of complexity or the scarcity of large-scale datasets for model calibration:

1.  The present state of global wetland modelling assumes generic wetlands without distinguishing their different types (Melton et al., 2013; Poulter et al., 2017). Like many other large-scale models of the current generation, WETMETH would not be appropriate for investigating the contribution from particular wetland types to regional or global $CH_4$ emissions (Aselmann and Crutzen, 1989).

2.  Since WETMETH is not based on a comprehensive soil biochemistry module and does not include the different pathways for $CH_4$ production and oxidation, the model is not suited for investigating the role of specific biological and chemical controls on wetland $CH_4$ emissions (Bridgham et al., 2013; Kwon et al., 2019).

3.  WETMETH does not simulate the contribution from wetland-specific vegetation species to $CH_4$ emissions, although some of these species can either lead to high emissions (e.g. sedges are vascular plants that can transport $CH_4$ through their aerenchyma) or low emissions (e.g. mosses are non-vascular plants that have been shown to develop a symbiotic relationship with methanotrophs) (Bridgham et al., 2013; Chen and Murrell, 2010).

4.  Ebullition and aerenchyma of vascular plants allow $CH_4$ produced in wetlands to escape to the atmosphere with little opportunity for oxidation (Segers, 1998; Whalen, 2005). Moreover, stems of woody trees are important conduits for $CH_4$ emissions in Amazonia, a major source region in the world (Pangala et al., 2017). By considering the net effect of all mechanisms transporting $CH_4$ to the atmosphere, WETMETH presents a limitation for investigating the relative contribution of transport mechanisms to $CH_4$ emissions across regions and at the global scale.

5.  Methane produced in northern wetlands can be stored underneath frozen soil during the winter and be released abruptly upon spring thaw (Mastepanov et al., 2013; Song et al., 2012). WETMETH does not currently feature such a storage of $CH_4$ in the soil column, which is probably more relevant for small-scale (sites) and short-term (days) than large-scale (regional) and long-term (seasonal) emissions (Fig. 9).

6.  As presented in this study, poorly constrained WETMETH parameters are tuned to estimates from northern high-latitude regions because large-scale datasets from other regions are scarce (see Section 4). A strong limitation comes with the assumption that the chosen parameter values are representative for $CH_4$ production and oxidation across the globe. However, the applied model calibration remains a reasonable approach given the scarcity of observations for wetland $CH_4$ production, oxidation, and emissions at the global scale.

Despite these limitations and the model simplicity, WETMETH is skillful when it comes to the simulation of mean seasonal, annual, and decadal wetland $CH_4$ emissions at the regional, hemispheric, and global scale (see Section 5.2). The implementation of WETMETH in a fully coupled ESM should advance research on the interactions between climate change and wetland $CH_4$ emissions in the context of global climate projections.



## 8 Conclusions

This paper introduces WETMETH – a process-based wetland $CH_4$ model developed for implementation in ESMs.
WETMETH is currently embedded in the UVic ESCM, a fully coupled EMIC. WETMETH is a computationally efficient model, applicable globally and, of appropriate complexity with respect to the current state of wetland $CH_4$ modelling. Unconstrained model parameters are tuned to potential $CH_4$ production rates from incubated soil samples and $CH_4$ emissions from northern wetlands due to the scarcity of large-scale datasets from other regions. Nevertheless, WETMETH reproduces well estimates of mean annual $CH_4$ emissions over the past few decades at the regional, hemispheric, and global scale.

Despite the importance of tropical wetlands in the global $CH_4$ budget (Kirschke et al., 2013; Saunois et al., 2016) and climate change (O'Connor et al., 2010; Zhang et al., 2017b), their areal extent and associated $CH_4$ emissions remain highly uncertain in both the literature and modelling work (including this study) due to a combination of limited ground-based measurements and process understanding (Pangala et al., 2017; Saunois et al., 2020; Sjögersten et al., 2014), as well as a low accuracy from remotely-sensed products especially over dense rainforests of Indonesia, Amazonia, and the Congo
River basin where new peatlands continue to be discovered to date (Dargie et al., 2017). Large-scale wetland mapping is a field of ongoing research (Tootchi et al., 2019) and further model development should focus on the improvement of wetland simulations in the tropics. In parallel, a compilation of tropical wetland $CH_4$ measurements from various sources into synthesis datasets would be beneficial for constraining wetland $CH_4$ processes in large-scale models.

The inclusion of wetland $CH_4$ processes in a fully coupled ESM allows to advance the research on the feedback
between climate change and wetland $CH_4$ emissions. The implementation of WETMETH in the UVic ESCM constitutes an ideal tool for investigating interactions between climate conditions and wetland $CH_4$ emissions from decadal to longer timescales. Of particular importance is the permafrost carbon feedback to climate change, in which $CH_4$ emissions from northern wetlands are expected to play an important role (Nzotungicimpaye and Zickfeld, 2017).

## Author contributions

CMN designed the research under the supervision of KZ, LFWL, JRM, and AHMD. LFWL contributed to the illustrated vertical profiles. CMN developed the wetland methane model with contributions from JRM and KZ. AHMD implemented the TOPMODEL approach in the UVic ESCM to which CMN applied a minor modification. CMN implemented the wetland methane model in the UVic ESCM with contributions from AHMD and ME. CMN performed the model calibration with contributions from CCT and KZ. CMN carried out the model simulations, evaluated the model performance, interpreted the results, and drafted the manuscript. All authors provided critical feedback on the manuscript and helped shape its final version.

## Competing interests

The authors declare that they have no conflict of interest.

## Code availability

The code for WETMETH 1.0 embedded in the University of Victoria Earth System Climate Model (UVic ESCM) version 2.9 used in this study is available at https://doi.org/10.5281/zenodo.4066112 (Nzotungicimpaye and Zickfeld, 2020).

## Data availability

WETMETH output variables analyzed in this study are archived at https://doi.org/10.20383/101.0215 and will be made accessible upon final publication of the manuscript.

## Acknowledgements

KZ and AHMD are each grateful for research funding from the National Sciences and Engineering Research Council of Canada (NSERC) Discovery Grants Program. The authors would like to thank the broad community of researchers who contributed to the datasets and model ensembles used in this study. We thank Catherine Prigent for sharing the GIEMS dataset, and Benjamin Poulter for sharing the SWAMPS-GLWD dataset and the GCP-CH4 model ensemble. We also thank Anthony Bloom, Jed Kaplan, and Olli Peltola for making their methane emission datasets (WetCHARTs ensemble, WETCHIMP ensemble, and the upscaled flux measurements, respectively) publicly available.





## Appendix A: Temperature-dependent $Q_{10}$ coefficient for methane production

Fig. A1 illustrates the different shapes of the temperature-dependency function for $CH_4$ production ($Q_{10}^{\frac{T_i - T_0}{10}}$; $T_0 = 273.15$

705    K) across a range of temperatures when considering: (i) a constant $Q_{10}$ of 4.2; and (ii) a temperature-dependent $Q_{10}$ coefficient given by $Q_{10}(T_i) = 1.7 + 2.5 \tanh [0.1 (T_{ref} - T_i)]$, where $T_{ref} = 308.15$ K. The temperature-dependent $Q_{10}(T_i)$ implies an optimal temperature for $CH_4$ production in WETMETH around 300.15 K. When $Q_{10}(T_i)$ decreases to reach negative values, its value in WETMETH is set to $10^{-3}$ to represent a very small methanogenic response to temperature changes (Fig. A1).

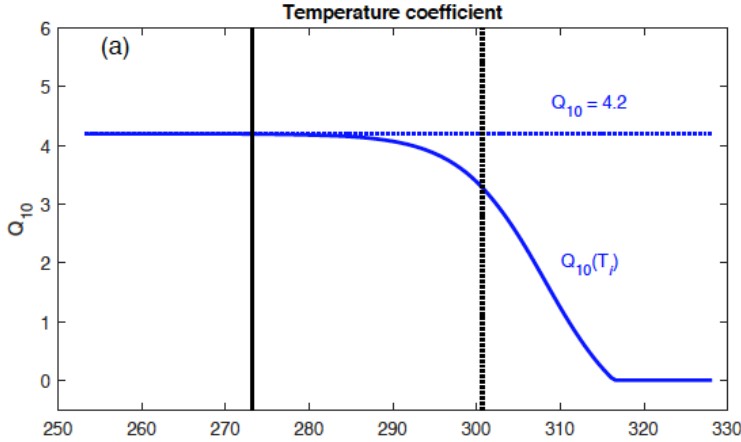

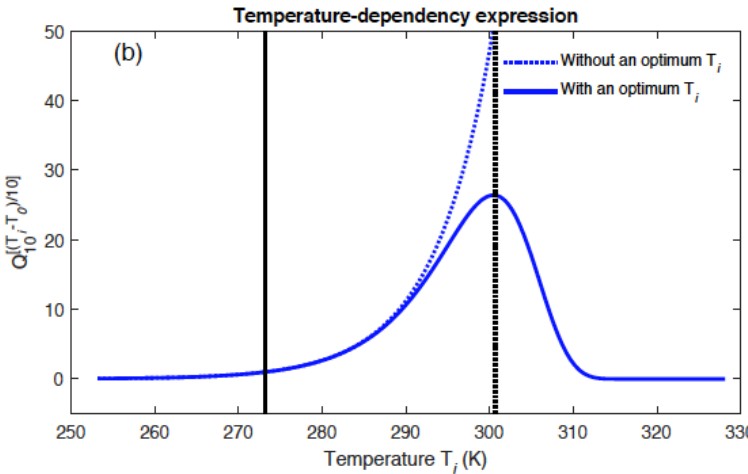

710

**Figure A1: (a) Differences between a constant $Q_{10}$ coefficient and a temperature-dependent $Q_{10}(T_i)$ coefficients and (b) implications for the temperature-dependency expression for $CH_4$ production ($Q_{10}[(T_i - T_0)/10]$). The temperature-dependent coefficient $Q_{10}(T_i) = 1.7 + 2.5 [\tanh(0.1 (308.15 - T_i))]$ allows to account for uncertainties in the $Q_{10}$ coefficient and to define an optimal temperature for $CH_4$ production around 300.15 K (dashed vertical line). The freezing point of water is shown at 273.15 K**
715    **(continuous vertical line).**





## Appendix B: Applied minor modification to the TOPMODEL approach

The TOPMODEL approach implemented in the UVic ESCM is based on the formulation by Gedney and Cox for global land surface models (Gedney and Cox, 2003). This approach combines the simulated hydrology with a prescribed topographic index to determine the occurrence of wetlands (surface inundation) and soil moisture heterogeneity at the sub-grid scale. The occurrence of wetlands is simulated in an area whose local topographic index ($\Lambda$) satisfies the following condition:

$$\Lambda_{min} \leq \Lambda \leq \Lambda_{max} , \tag{B1}$$

where $\Lambda_{min}$ is a lower threshold that can be related to under-saturation conditions and $\Lambda_{max}$ is an upper threshold that can be related to over-saturation conditions.

In the initial work by Gedney and Cox, $\Lambda_{min}$ depends on the transmissivity of the entire soil column (T(0)), the transmissivity of the soil column below the mean water table depth ($z_w$) of the grid box (T($z_w$)) as well as the mean topographic index ($\Lambda_{mean}$). It is calculated as $\Lambda_{min} = \ln\frac{T(0)}{T(z_w)} + \Lambda_{mean}$. While $\Lambda_{mean}$ is static and prescribed with a topographic index map, both transmissivities (T(0) and T($z_w$)) are simulated and non-static for a specific grid cell. Hence, $\Lambda_{min}$ is a non-static and grid-dependent threshold. Unlike $\Lambda_{min}$, $\Lambda_{max}$ is a static and global threshold. This threshold is applied to constrain the occurrence of wetlands in areas of stagnant water based on the assumption that locations where the water table rises well above the surface would be characterized by streamflow.

For the current study, a minor modification is applied to the above TOPMODEL approach. The revision consists of using a non-static and grid-dependent $\Lambda_{max}$ instead of a static and global threshold. Following the formulation by Comyn-Platt and colleagues (Comyn-Platt et al., 2018), an expression for $\Lambda_{max}$ that depends on $\Lambda_{min}$ is currently used in the UVic ESCM. This threshold is defined as:

$$\Lambda_{max} = \Lambda_{min} + \Lambda_{range} , \tag{B2}$$

where $\Lambda_{range}$ is a global tuning parameter ($\Lambda_{range}= 0.93$ in the version of the UVic ESCM used in this study).

In summary, unlike the initial work by Gedney and Cox (Gedney and Cox, 2003), the modified TOPMODEL approach considers two non-static and grid-dependent thresholds ($\Lambda_{min}$ and $\Lambda_{max}$) for the identification of wetlands across the globe.

## Appendix C: Unit conversion for potential methane production rates

Here, we describe steps followed for converting units of maximum $CH_4$ production rates measured in laboratory incubations from a soil weight basis ($\mu$g C g DW$^{-1}$ hr$^{-1}$) to a soil volume basis (kg C m$^{-3}$ s$^{-1}$). This unit conversion relies on the soil bulk density (BD in g cm$^{-3}$) from the site of origin. The following two steps illustrate the applied unit conversion. In the first step, the potential $CH_4$ production rates ($P_{d,0}$) are converted from $\mu$g C g DW$^{-1}$ hr$^{-1}$ to $\mu$g C cm$^{-3}$ hr$^{-1}$ as follows:

$$P_{d,1} = (BD)\, P_{d,0} \tag{C1}$$

Then, the conversion of $P_{d,1}$ from $\mu$g C cm$^{-3}$ hr$^{-1}$ to kg C m$^{-3}$ s$^{-1}$ is done as follows:





$$P_{d,2} = \frac{\delta}{\gamma} P_{d,1}, \tag{C2}$$

where $\delta$ encompasses the conversion factors from $\mu g$ to kg and from $cm^{-3}$ to $m^{-3}$ ($\delta = 10^{-3} kg\ m^{-3}$); and $\gamma$ is the number of seconds per hour ($\gamma = 3600\ s$).



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
