# Peer review of "WETMETH 1.0: A new wetland methane model for implementation in Earth system models"

_Geoscientific Model Development, 2020_

## Referee Comment (RC1) · Anonymous Referee #1 · 18 Nov 2020

Summary:

The WETMETH model represents some advances in modelling science, although one of its "novel" components (vertical methane production) is also in other models. Even though WETMETH is to be included in an Earth system model, its detailed validation is highly skewed to the middle and high latitudes. More detailed validation needs to be undertaken in tropical regions. The scheme also needs a more thorough justification and calibration of its methane production optimal temperature formulation.

The manuscript is generally well-written and constructed, with appropriate figures and tables.

[Figure]

Model design, validation and calibration:

The manuscript describes the three pathways by which methane is produced by methanogens in the soil, with the main pathways utilising either acetate or CO2. Root exudates provide a large source of acetate, but it is unclear from the WETMETH model description that any of the soil carbon pools include root exudates. If root exudates are included in a soil pool, does this pool have a fast enough turnover time appropriate for their decomposition, or is it combined with other compounds which have slower decomposition rates?

An optimal temperature for methane production is included in WETMETH, with the model being particularly sensitive to the value it is assigned. The formulation used is taken from Wu et al 2016, which is calibrated using data from mid and high latitudes bogs and fens. It is also based on the optimal temperature for soil CO2 production.

Methane production via the acetoclastic pathway is not dependent on CO2 availability however. So is it valid to use the same optimum temperature for methane production? Also is this optimum temperature value used equally valid in the tropics where one might expect soil microbes to be acclimatized to higher temperatures?

The model produces very different latitudinal patterns in flux per unit area to other estimates (Figure 11b, page 24), which could be due to the optimum temperature. This is especially the case over the tropics where the authors acknowledge that the overestimation of wetland area may be compensating for the low flux intensity, resulting in an overall flux which is more comparable to other estimates. This optimal temperature parameterization and the value chosen needs more justification, analysis and discussion.

The detailed validation of WETMETH is skewed towards the extra-tropics. There is insufficient validation against available tropical methane measurements. The assessment of tropical fluxes is made by comparison against other models from (climate model) grid to regional scales. This issue is highlighted by the fact that WETMETH

produces very different flux rates per unit area in the tropics relative to other models (see above). Adequate validation over the tropics is of particular importance because tropical wetland methane emissions dominate the global wetland budget, and WET-METH is intended for use in Earth system models.

Specific comment:

The manuscript states: "....particularity of WETMETH among relatively simple models is that the model accounts for...... a depth-dependent representation for methane production" (page 27 line 620). The addition of a vertical dependent methane production is not entirely novel within models of similar complexity (e.g. Comyn-Platt et al 2018).

———————————————

---

## Referee Comment (RC2) · Anonymous Referee #2 · 10 Jan 2021

The paper describes an intermediate complexity approach to represent methane production and oxidation in an Earth system model. Modeling the two component fluxes of methane from wetlands is challenging and this paper is a good contribution to the existing land surface models engaged in wetland methane emission research. The paper is very well written and is excellently referenced.

I have just minor comments and clarifications:

In the introduction, please mention 'acetotrophic methanogenesis' and 'hydrogenotrophic methanogenesis' when discussing the three pathways for methane production from microbes.

[Figure]

Fig 1 – suggest adding a CH4 production figure

Eq 1 – is Ci representative of total soil C, or more mobile source of C, like the intermediate pool (if using a CENTURY type model) ?

Eq 1 – specify that S(theta) is for vertical profile, not for the entire grid fraction

Is there any coupling of permafrost sub routines w your CH4 production scheme? Do they both use same soil temperature – and so if soil temperature is cold, then no CH4 emissions?

What is the model timestep of CH4 production and oxidation?

Does soil carbon get reduced to account for carbon lost through CH4 production?

Is Zoatz the same as Zoxic, I was a confused here what the difference is.

Ultimately, the oxidation rate is fixed through space and time, my calculation is that 96% of gross methane production is oxidized, e.g., 1-exp(-(0.05/0.0146)) – is this correct?

Why not vary Zoxic to represent a dynamic water table height? I didn't understand why this is fixed at 0.05 globally and over time.

---

## Author Comment (AC1) · 5 Jun 2021

**General information**

Please consider the following information that we intend to add in the submitted manuscript if we are invited to submit a revised version:

- Figure 11b with a revised curve for WETCHIMP emission intensities after double-checking zonally-averaged wetland areas from each model.
- New references on the optimal temperature for soil microbial processes and in particular CH4 production in wetlands: (i) Schipper et al. (2014), Global Change Biology, 20, 3578-3586; (ii) Metje and Frenzel (2007), Environmental Microbiology, 9, 954-964.

Reviewer comments are reproduced in plain text and responses given in bold.

**Response to reviewer #1**

Summary:

The WETMETH model represents some advances in modelling science, although one of its "novel" components (vertical methane production) is also in other models. Even though WETMETH is to be included in an Earth system model, its detailed validation is highly skewed to the middle and high latitudes. More detailed validation needs to be undertaken in tropical regions. The scheme also needs a more thorough justification and calibration of its methane production optimal temperature formulation. The manuscript is generally well-written and constructed, with appropriate figures and tables.

**Thanks for the positive and constructive comments on the manuscript. Below we provide our responses to each comment.**

Model design, validation and calibration:

The manuscript describes the three pathways by which methane is produced by methanogens in the soil, with the main pathways utilising either acetate or CO2. Root exudates provide a large source of acetate, but it is unclear from the WETMETH model description that any of the soil carbon pools include root exudates. If root exudates are included in a soil pool, does this pool have a fast enough turnover time appropriate for their decomposition, or is it combined with other compounds which have slower decomposition rates?

WETMETH does not distinguish root exudates from other organic matter compounds in the soil. That is, (1) WETMETH considers the soil carbon pool as a single pool aggregating all sources of organic matter substrates (e.g. litter fall and root exudates), and (2) there is currently no distinction between fast and slow pathways of CH4 production in our model. We assume that rates of CH4 production in WETMETH encompass all possible CH4 production pathways when combined together.

To clarify these points, we will include the following statements in Section 3.1.1:

• First paragraph (statement to be included after line 187 of the original text) -> We consider Ci to be the aggregate of all sources of soil carbon (i.e. organic matter) such as litter-fall and root exudates.

• Second paragraph (statement to be included after line 195 of the original text) -> In this first version of WETMETH, we combine all possible pathways for CH₄ production in wetlands (see Section 2.1) without distinguishing fast and slow pathways.

An optimal temperature for methane production is included in WETMETH, with the model being particularly sensitive to the value it is assigned. The formulation used is taken from Wu et al 2016, which is calibrated using data from mid and high latitudes bogs and fens. It is also based on the optimal temperature for soil CO2 production. Methane production via the acetoclastic pathway is not dependent on CO2 availability however. So is it valid to use the same optimum temperature for methane production? Also is this optimum temperature value used equally valid in the tropics where one might expect soil microbes to be acclimatized to higher temperatures?

**Optimum temperature for CH4 production versus CO2 production and availability**

This is an interesting point linking pathways of CH4 production in wetlands to soil CO2 production and CO2 availability. While it is possible that there exist different optimal temperatures for CH4 production via the acetoclastic pathway versus other methanogenesis pathways, there is a paucity of literature on these optimal temperatures for the various methanogenesis pathways. For this reason, we use one formulation for the optimal temperature for CH4 production in our initial development of WETMETH.

**Same optimum temperature for CH4 production across the globe**

We were not able to find specific optimal temperatures for CH4 production in different climate zones. Hence, for our first implementation of WETMETH, we chose to apply a constant optimum temperature for CH4 production based on published studies, which is mainly based on data from northern high-latitude environments (e.g. Dunfield et al. 1993; Metje and Frenzel 2007). The default optimum temperature for CH4 production in WETMETH (~27°C) is subject to a sensitivity analysis (See Section 6).

Reference to Wu et al. (2016) for our formulation of the optimal temperature for CH4 production Due to this comment, we realize the way we cite Wu et al. (2016) with regard to the Q10 coefficient and optimal temperature formulation may not be clear. The form of the equation used for the Q10 coefficient in WETMETH is analogous to the one used in Wu et al. (2016), but the coefficients in the equation differ (our mathematical expression for the CH4 production is Q10(Ti) =  $1.7 + 2.5 \tanh[(0.1$ (308.15-Ti))], whereas that used in Wu et al. (2016) for soil CO2 production and respiration is Q10(Ti) =  $1.44 + 0.56 \tanh[0.075 (46-Ti)]).$

To avoid further confusions, we will revise the related statement in Section 3.1.1 of the submitted manuscript (lines 207-208) as follows:

• "This mathematical formulation is analogous to one used for soil respiration in another study (Wu et al., 2016)".

The model produces very different latitudinal patterns in flux per unit area to other estimates (Figure 11b, page 24), which could be due to the optimum temperature. This is especially the case over the tropics where the authors acknowledge that the overestimation of wetland area may be compensating for the low flux intensity, resulting in an overall flux which is more comparable to other estimates. This optimal temperature

parameterization and the value chosen needs more justification, analysis and discussion.

**Justification of the optimal temperature parameterization**

The existence of an optimum temperature for CH4 production in wetlands is relatively well established in the literature (e.g. Dunfield et al. 1993; Metje and Frenzel, 2007; Dean et al. 2018). As indicated in Appendix A1, the chosen value for this optimum temperature in WETMETH (~27°C) is within the range of available estimates (i.e. 25-30°C). This default value is subject to a sensitivity analysis (through  $T_{ref}$ ) in Section 6.

To clarify these points, we will revise the following statement in Section 3.1.1 of the revised manuscript (with new text highlighted in yellow):

• "In order to account for ... the occurrence of an optimal temperature for CH4 production (Blake et al., 2015; Dean et al., 2018; Dunfield et al., 1993), a temperature-dependent  $Q_{10}$  is considered in WETMETH. Its mathematical formulation is  $Q_{10}(T_i) = 1.7 + 2.5tanh$  [0.1  $(T_{ref}-T_i)$ ], where  $T_{ref} = 308.15$  K is a reference temperature that is used to define an optimal temperature for CH4 production (Table 1). This formulation is defined in analogy to a mathematical expression used for soil respiration in another study (Wu et al., 2016), and it enables to account for an optimal temperature for CH4 production of ~300.15 K (i.e. 27°C) which is consistent with previous studies (Dunfield et al., 1993; Metje and Frenzel, 2007). Additional information on this formulation and its implications for the temperature-dependency of CH4 production are provided in Appendix A1".

In the above paragraph, (i) we clarify that the optimal temperature for CH4 production is defined (or parameterized) through  $T_{ref}$  in the Q10 formulation, and (ii) we justify the default value chosen in WETMETH for this optimal temperature.

**Analysis with regards to the optimal temperature parameterization**

In our model, we parameterize the optimal temperature for CH4 production through  $T_{ref}$  in the Q10 formula described in Section 3.1.1. (third paragraph) as well as Appendix A1. Given that the optimal temperature for CH4 production is uncertain, we conducted a sensitivity analysis on the default value for the optimal temperature in WETMETH (i.e. ~27°C) which is defined through  $T_{ref}$  in the Q10 formula. In Section 6 (see third paragraph), we show that simulated CH4 emissions from tropical wetlands are more sensitive to changes in  $T_{ref}$  (therefore changes in the optimal temperature value for CH4 production) than CH4 emissions from northern wetlands. This result suggests that, in the present-day climate, wetland CH4 emissions in the tropics are more dependent on the optimal temperature for CH4 production than wetland CH4 emissions in the boreal and Arctic regions.

**Discussion**

We plan to include a new bullet point (see below) in Section 7 discussing the optimal temperature for CH4 production in the context of this manuscript and the modelling of wetland CH4 emissions.

 "While the existence of an optimal temperature for CH4 production in wetlands is relatively well established in the literature (Dean et al., 2018), there are currently no estimates of such an optimal temperature for different climate zones across the globe. Previous studies suggest a range of 25-30°C for such an optimal temperature based on measurements of CH4 production in northern wetlands (Dunfield et al., 1993; Metje and Frenzel, 2007). In WETMETH, we use a global value for this optimal temperature (~27°C) which is assumed to be valid for CH4 production in both tropical and extra-tropical wetlands (see Section 3.1.1 and Appendix A1). However, our sensitivity analysis suggests that, in the present-day climate, wetland CH4 emissions in the tropics are more dependent on the optimal temperature for CH4 production than wetland CH4 emissions in the boreal and Arctic regions (see Section 6). The optimal temperature for CH4 production in WETMETH, along with other factors such as areal wetland extents, contributes to inter-model differences in simulated wetland CH4 intensities in the tropics (see Figure 11b)".

The detailed validation of WETMETH is skewed towards the extra-tropics. There is insufficient validation against available tropical methane measurements. The assessment of tropical fluxes is made by comparison against other models from (climate model) grid to regional scales. This issue is highlighted by the fact that WETMETH produces very different flux rates per unit area in the tropics relative to other models (see above). Adequate validation over the tropics is of particular importance because tropical wetland methane emissions dominate the global wetland budget, and WETMETH is intended for use in Earth system models.

We agree that adequate evaluation in the tropics is needed, however we are limited by the scarcity of large-scale observations across the tropics; a challenge shared by the wide community of scientists working on large-scale CH4 emissions: from those contributing to observation syntheses and database development (e.g. FLUXNET-CH4 by Knox et al. 2019) to those putting together CH4 budget reports (e.g. Saunois et al. 2020). As we state at the beginning of Section 5.2, our model validation focuses on large-scale wetland CH4 emissions given the coarse grid resolution of the UVic ESCM into which WETMETH has been embedded, and thus we require large-scale observations which, to our knowledge, are not available.

Nevertheless, we show (in Table 2) that tropical wetland CH4 emissions simulated by WETMETH fall in the range of estimates available at the large spatial scales needed for an appropriate comparision - namely upscaled measurements (Sjörgesten et al. 2014) as well as results from process-based models (Melton et al. 2013) - although we are aware that these estimates are poorly constrained.

Furthermore, we acknowledge the particular importance of tropical wetland CH4 emissions in the global CH4 budget and suggest that more observations and improved process-understanding are needed to constrain these CH4 emissions (see Section 8 - Conclusions, second paragraph).

Finally, we also acknowledge (e.g. in Section 5.1 & Section 8) that tropical wetlands simulated by the UVic ESCM are generally more widespread than predicted by observation-based datasets, implying the need for future work to improve the spatial distribution of these wetlands in the UVic ESCM. In the advent of more reliable data products for large-scale wetland CH4 emissions in the tropics in the future, along with improved tropical wetlands in the UVic ESCM, we expect that further model development will consider detailed model validation with respect to tropical wetland CH4 emissions.

**Specific comment:**

The manuscript states: ": : :.particularity of WETMETH among relatively simple models is that the model accounts for: : :: : : a depth-dependent representation for methane production" (page 27 line 620). The addition of a vertical dependent methane production is not entirely novel within models of similar complexity (e.g. Comyn-Platt et al 2018). Our intention with that statement (in the last paragraph on page 27) is to highlight the novelty with respect to calibrating model parameters for depth-dependent CH4 production against measurements from laboratory incubations. To the best of our knowledge, such a calibration for CH4 production has never been done in other global wetland CH4 models.

For better clarity, we intend to rephrase our statement as follows:

"The particularity of WETMETH among relatively simple models is that the model accounts for ..., a calibration of depth-dependent CH4 production rates against potential CH4 production rates from laboratory incubations, dynamic CH4 oxidation ...".

**Response to reviewer #2**

The paper describes an intermediate complexity approach to represent methane production and oxidation in an Earth system model. Modeling the two component fluxes of methane from wetlands is challenging and this paper is a good contribution to the existing land surface models engaged in wetland methane emission research. The paper is very well written and is excellently referenced.

I have just minor comments and clarifications:

**We thank the reviewer for the positive and constructive comments on the manuscript. Please see below our responses to each of their comments and requests for clarifications.**

In the introduction, please mention 'acetotrophic methanogenesis' and 'hydrogenotrophic methanogenesis' when discussing the three pathways for methane production from microbes.

Thank you for this suggestion. Following your recommendation, we will mention the terminologies associated with the three methanogenesis pathways (acetotrophic methanogenesis, hydrogenotrophic methanogenesis, and methylotrophic methanogenesis) in the revised manuscript.

Fig 1 – suggest adding a CH4 production figure

The reason for depicting CH4 concentration rather than production in Figure 1 is that concentration is what is typically measured in the vast majority of field observations. As a simplification for the purpose of this illustration, the CH4 concentration curve in Figure 1 was assumed to mirror CH4 production rates at depth within the soil system. The reason for assuming such correspondence (as explained in the text of the paper) is that the main effect on production rates is likely to be % organic content under anaerobic conditions, but under aerobic conditions, the main effect is more likely to be a balance between organic content versus increasing availability of O2 that could support aerobic decomposer communities that compete for the available organic carbon substrate. Under the aerobic scenario, we assumed CH4 oxidation would also contribute to lower CH4 concentrations in the upper soil, but that much of this oxidation would be based on consuming CH4 produced deeper in the soil that is diffusing upward into and through the non-saturated layer.

For clarification, the following sentence will be added to the figure caption:

• "As an illustrative simplification, the CH4 concentration profile is assumed to mirror CH4 production rates at depth within the soil column (see explanatory text in Section 2.4)".

Eq 1 – is Ci representative of total soil C, or more mobile source of C, like the intermediate pool (if using a CENTURY type model) ?

**Ci represents total soil C in a given ground layer *i* (please see the first statement of Section 3.1.1 from the original text as reproduced in our next response).**

Eq 1 – specify that S(theta) is for vertical profile, not for the entire grid fraction Is there any coupling of permafrost sub routines w your CH4 production scheme? Do they both use same soil temperature – and so if soil temperature is cold, then no CH4 emissions?

**S(theta)**

In the first paragraph of Section 3.1.1 of the submitted manuscript (reproduced below), we indicated that CH4 production in our model (as described by Equation 1) is calculated in individual soil layers whereby  $S(\theta_i$  characterizes soil moisture saturation in a given soil layer *i*, and not for the entire grid fraction.

"For any land location, the rate of CH4 production in an underlying soil layer *i* ( $P_i$  in kg C m-3 s-1) is parameterized as:

 $P_{i} = S(\theta_{i}) C_{i} r Q_{10}^{\frac{T_{i} - T_{0}}{10}} \exp(-\frac{z_{i}}{\tau_{\text{prod}}}),$

(1)

where  $S(\theta_i)$  is the fraction of the soil layer that is saturated with water, and  $C_i$  is the amount of soil carbon (in kg C m-3) in the layer".

**Permafrost sub routines & CH4 production scheme**

Our CH4 production scheme is coupled to the permafrost sub routines in the UVic ESCM and they both use the same soil temperature. In the UVic ESCM-WETMETH coupling, we assume that:

- 1. When a given soil layer *i* is frozen ( $T_i < 0$ ), CH4 production shuts down in that layer;
- 2. CH4 production would still occur in unfrozen soil layer ( $T_i > 0$ ) contingent to other factors (e.g. soil carbon content and moisture saturation).

The above assumptions are described in Section 3.1.1 and reflected in Equation 1. A direct implication of these assumptions is that no CH4 production and emissions occur when the entire soil column is frozen. We acknowledge that any bias in soil temperature (as simulated by the UVic ESCM) will be reflected in CH4 production and emission rates.

**What is the model timestep of CH4 production and oxidation?**

CH4 production and oxidation in our model (that is, the UVic ESCM-WETMETH coupling) are calculated every 5 days and integrated with a 30-day timestep. This 30-day timestep integration is similar to that of soil respiration and other terrestrial carbon fluxes defined in the dynamic vegetation model (TRIFFID) component of the UVic ESCM.

To clarify this point, we will add a statement (see below) on this model timestep for all terrestrial carbon fluxes in the UVic ESCM through TRIFFID. This statement will be added in Section 3.2 (third paragraph):

• "Through TRIFFID, all terrestrial carbon fluxes in the UVic ESCM are integrated with a 30-day timestep (Meissner et al., 2003)".

Does soil carbon get reduced to account for carbon lost through CH4 production? No, such a reduction of soil carbon is not implemented in the model presented in the manuscript. However, the reduction of soil carbon following CH4 production is considered in further model developments with the global CH4 cycle in order to account for the conservation of carbon in the Earth system. This effect is negligible in the historical simulation of wetland CH4 emissions and the climate system.

Is Zoatz the same as Zoxic, I was a confused here what the difference is.

Zoxic and Zoatz are different: Zoxic represents the oxic zone depth (illustrated with the orange twosided arrows in Figure 2), whereas Zoatz represents the thickness of the oxic-anoxic transition zone to account for the availability of oxygen in the top layer saturated with water (i.e. top of the anoxic zone). The bottom of Zoatz defines the oxic-anoxic interface (illustrated with the red dashed horizontal lines in Figure 2).

**Overall:**

- 1. Zoatz is included in Zoxic;
- 2. The two terms are only identical when the soil surface is inundated with water (e.g. Figure 2a).

To clarify these points, we will include the following statement in the revised manuscript:

• "When the soil surface is inundated,  $z_{oatz}$  is identical to  $z_{oxic}$  (Fig. 2a). Otherwise,  $z_{oatz}$  is only a fraction of  $z_{oxic}$  (Fig. 2b)".

Ultimately, the oxidation rate is fixed through space and time, my calculation is that 96% of gross methane production is oxidized, e.g., 1-exp(-(0.05/0.0146)) – is this correct? Why not vary Zoxic to represent a dynamic water table height? I didn't understand why this is fixed at 0.05 globally and over time.

In our model, CH4 oxidation does vary with a dynamic Zoxic as illustrated by Figure 2 (see panel a versus panel b). Your calculation is correct for a case when the soil is inundated (i.e. for Zoxic = Zoatz = 0.05 m, 96.7% of CH4 produced in the soil column gets oxidized in transit to emission). However, when the soil is not inundated, the fraction of produced CH4 that gets oxidized before release (1) is greater than 97%; and (2) increases as Zoxic deepens.

To clarify this point, we will include the following sentence in the revised manuscript (Section 4.2):

• "This tuned value for  $\tau_{oxid}$  is listed in Table 1 and implies that ~97% of the CH4 produced in the soil column gets oxidized when the soil surface is inundated".

Please note that we assume a static Zoatz (0.05 m) in this first version of WETMETH (as stated in Section 3.1.2, second paragraph). Further model development could explore how to represent Zoatz in a dynamic way.